# Measurements from the *RV Ronald H. Brown* and related platforms as part of the Atlantic Tradewind Ocean-Atmosphere Mesoscale Interaction Campaign (ATOMIC)

Patricia K. Quinn[1], Elizabeth J. Thompson[2], Derek J. Coffman[1], Sunil Baidar[3,4], Ludovic Bariteau[2], Timothy S. Bates[1,5], Sebastien Bigorre[6], Alan Brewer[4], Gijs de Boer[2,3], Simon P. de Szoeke[7], Kyla Drushka[8], Gregory R. Foltz[9], Janet Intrieri[2], Suneil Iyer[8], Chris W. Fairall[2], Cassandra J. Gaston[10], Friedhelm Jansen[11], James E. Johnson[1,5], Ovid O. Krüger[12], Richard D. Marchbanks[3,4], Kenneth P. Moran[2,3], David Noone[13], Sergio Pezoa[2], Robert Pincus[2,3], Albert J. Plueddemann[6], Mira L. Pöhlker[12], Ulrich Pöschl[12], Estefania Quinones Melendez[7], Haley M. Royer[10], Malgorzata Szczodrak[10], Jim Thomson[8], Lucia M. Upchurch[1,5], Chidong Zhang[1], Dongxiao Zhang[1,5], and Paquita Zuidema[10]

[1]NOAA Pacific Marine Environmental Laboratory (PMEL), Seattle, WA, USA
[2]NOAA Physical Sciences Laboratory (PSL), Boulder, CO, USA
[3]Cooperative Institute for Research in Environmental Sciences (CIRES), University of Colorado, Boulder, CO
[4]NOAA Chemical Sciences Laboratory (CSL), Boulder, CO, USA
[5]Cooperative Institute for Climate Ocean and Ecosystem Studies (CICOES), University of Washington, Seattle, WA, USA
[6]Woods Hole Oceanographic Institution (WHOI), Woods Hole, MA, USA
[7]Oregon State University, Corvallis, OR, USA
[8]University of Washington, Applied Physics Laboratory (APL), Seattle, WA
[9]NOAA Atlantic Oceanographic and Meteorological Laboratory (AOML), Miami, FL, USA
[10]Rosenstiel School of Marine and Atmospheric Science, University of Miami, Miami, FL, USA
[11]Max Planck Institute for Meteorology, Hamburg, Germany
[12]Max Planck Institute for Chemistry, Mainz, Germany
[13]University of Auckland, Auckland, NZ

*Correspondence:* Patricia K. Quinn (patricia.k.quinn@noaa.gov)

**Abstract.** The Atlantic Tradewind Ocean-Atmosphere Mesoscale Interaction Campaign (ATOMIC) took place from January 7 to July 11, 2020 in the tropical North Atlantic between the eastern edge of Barbados and 51°W, the longitude of the Northwest Tropical Atlantic Station (NTAS) mooring. Measurements were made to gather information on shallow atmospheric convection, the effects of aerosols and clouds on the ocean surface energy budget, and mesoscale oceanic processes. Multiple platforms were deployed during ATOMIC including the *NOAA RV Ronald H. Brown* (RHB) (Jan. 7 to Feb. 13) and WP-3D Orion (P-3) aircraft (Jan. 17 to Feb. 10), the University of Colorado's Robust Autonomous Aerial Vehicle – Endurant and Nimble (RAAVEN) Uncrewed Aerial System (UAS) (Jan. 24 to Feb. 15), NOAA- and NASA-sponsored Saildrones (Jan. 12 to Jul. 11), and Surface Velocity Program Salinity (SVPS) surface ocean drifters (Jan. 23 to Apr. 29). The *RV Ronald H. Brown* conducted *in situ* and remote sensing measurements of oceanic and atmospheric properties with an emphasis on mesoscale oceanic-atmospheric coupling and aerosol-cloud interactions. In addition, the ship served as a launching pad for Wave Gliders, Surface Wave Instrument Floats with Tracking (SWIFTs), and radiosondes. Details of measurements made from the *RV Ronald H. Brown*, ship-deployed assets, and other platforms closely coordinated with the ship during ATOMIC are provided here. These platforms include Saildrone 1064 and the RAAVEN UAS as well as the

Barbados Cloud Observatory (BCO) and Barbados Atmospheric Chemistry Observatory (BACO). Inter-platform
comparisons are presented to assess consistency in the data sets. Data sets from the *RV Ronald H. Brown* and
deployed assets have been quality controlled and are publicly available at the NOAA Physical Sciences Laboratory
(PSL) ATOMIC ftp server (ftp://ftp2.psl.noaa.gov/Projects/ATOMIC/data/). In addition, the data have been
submitted to NOAA's National Centers for Environmental Information (NCEI) data archive
(https://www.ncei.noaa.gov/) for Digital Object Identifiers (DOIs). Point of contact information and links to
individual data sets are provided herein.

**1. Introduction**

Shallow, liquid clouds persist at altitudes of hundreds to a few thousand meters above most of the world's oceans.
Convection and mixing in the boundary layer can lead to the formation of shallow clouds, which can drive more
mixing throughout the cloud layer and result in deeper convection. These clouds reflect incoming solar radiation and
lead to a cooling of the surface (Vial et al., 2016). In addition, shallow mixing influences sea surface temperature
(SST) and salinity by moderating the air-sea exchanges of energy and moisture (Stevens et al., 2016). Climate
models have difficulty accurately representing low clouds in trade-wind regions because many of the processes
involved in their formation occur at sub-grid scales (Bony et al., 2015). Improving model performance requires
measurements that will result in a better understanding of 1) the boundary layer conditions that lead to cloudiness, 2)
the influence of clouds and the atmospheric boundary layer on the upper ocean mixed layer, and in turn, 3) the
influence of ocean mixing processes on surface fluxes and the atmospheric boundary layer.

ATOMIC took place in the boreal winter to study shallow convection and low, liquid clouds at a time of year when
other cloud types are mostly absent. ATOMIC is the U.S. complement to the Elucidating the Role of Clouds
Circulation Coupling in Climate Campaign (EUREC[4]A) (Bony et al., 2017;Stevens et al., 2020). Together,
ATOMIC and EUREC[4]A involved 4 research vessels, 4 research aircraft, land-based observations from Barbados,
and uncrewed sea-going and aerial vehicles. The ATOMIC - EUREC[4]A study region stretched from the eastern
shores of Barbados to the NTAS buoy located ~500 NM to the northeast and south along the coast of South America
to ~ 5°N. EUREC[4]A platforms focused on the western portion of the study area while the *RV Ronald H. Brown* and
P-3 aircraft worked primarily in the eastern, upwind sector from mid-January to mid-February (Fig. 1). ATOMIC
was composed of two legs with Leg 1 conducted between Jan. 7 and 25, 2020 and Leg 2 conducted between Jan. 28
and Feb.13, 2020. NOAA- and NASA-sponsored Saildrones covered the entire study area between January and July
2020. The RAAVEN UAS flew near shore from Morgan Lewis on the eastern side of Barbados between January 24
and February 15. SVPS type surface ocean drifters were deployed from the *RV L'Atalante* and operated along the
South American coast (Jan. 23 to Apr. 29).


**Figure 1**. Tracks of the *RV Ronald H. Brown,* Wave Gliders, and SWIFTS during ATOMIC (colored by seawater
skin T calculated by PSL) for a) Leg 1 and b) Leg 2. Tracks for Wave Gliders, and SWIFTS are magnified in c) Leg
1 and d) Leg 2. The portion of the EUREC⁴A study area overlapping with ATOMIC is indicated by the solid green
line in b). Locations of *RV Ronald H. Brown* stations, MOVE, and BCO/BACO are also shown in a) and b).

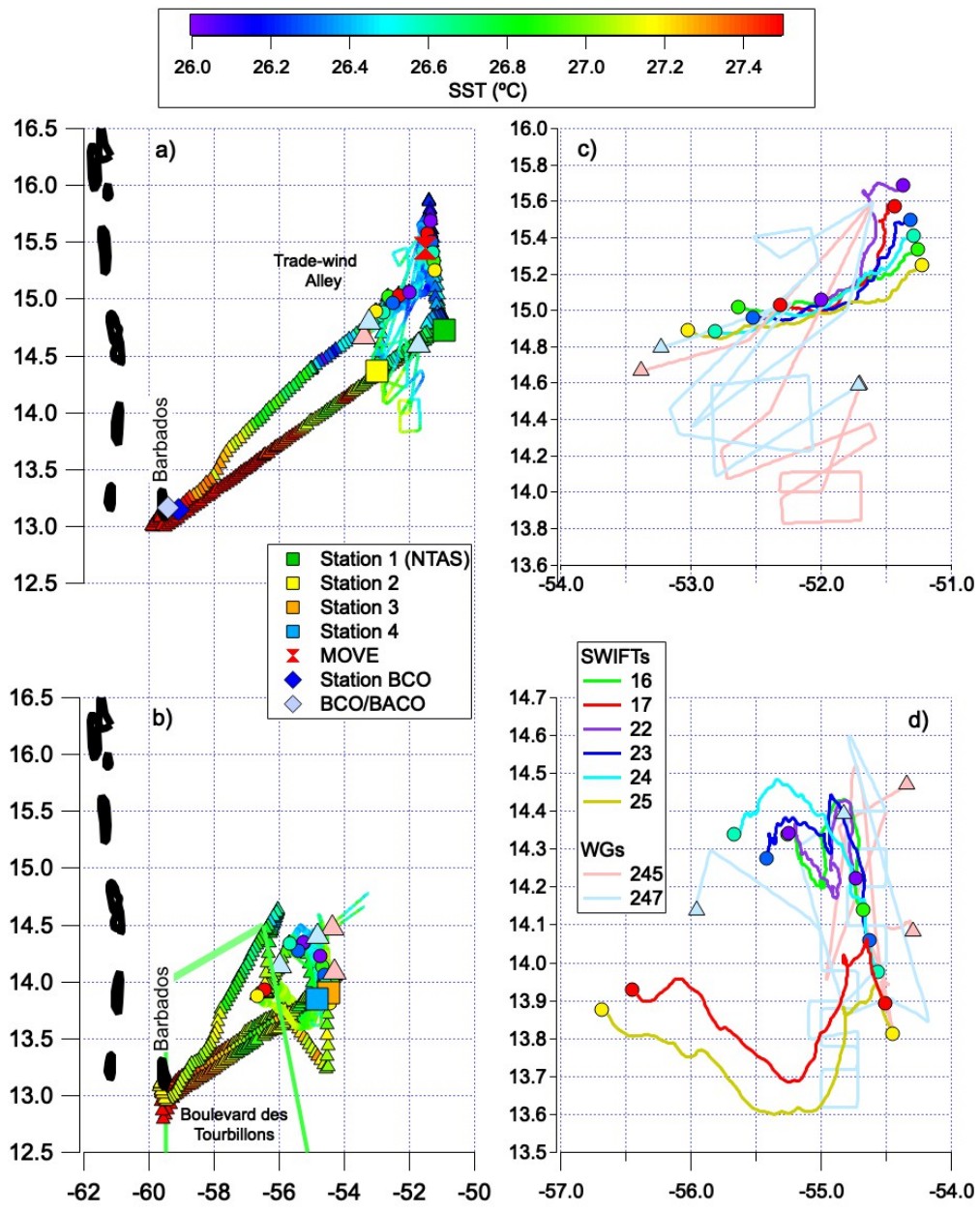

A thorough description of the objectives of ATOMIC and first highlights of the data analyses are presented in
Zuidema (2020). A description of data collected from the P-3 is described in Pincus et al. (2020) and data collected
by the RAAVEN are documented in de Boer et al. (2020). Here, a detailed overview of the data collected from the
*RV Ronald H. Brown* and deployed assets is provided. The goal is to document the sampling strategy,
instrumentation used, and data availability to advance the widespread use of the data by the ATOMIC and broader
research communities. A description of the sampling strategy, including coordination with other platforms, is
described in Sect. 2. Also detailed in Sect. 2 are the measurements made from the *RV Ronald H. Brown,* the NTAS
moored buoy, Wave Gliders and SWIFT vessels, Saildrones, RAAVEN UAS, and SVPS drifters. An overview of
oceanic and atmospheric conditions sampled is provided in Sect. 3. Results from inter-platform comparisons of
atmospheric and oceanic parameters are detailed in Sect. 4. Data availability, format, and quality control are
described in Sect. 5 along with links to individual data sets. Available measurement uncertainties are reported in the
data set metadata at ftp://ftp2.psl.noaa.gov/Projects/ATOMIC/data/ and https://www.ncei.noaa.gov/.

## 2. Sampling strategy and measurements

Sampling onboard the *RV Ronald H. Brown* took place from Jan. 7 to Feb. 13, 2020 and focused on the region
between 57°W and 51°W east of Barbados and between 13° and 16°N in the so-called Trade Wind Alley (Fig. 1).
The overarching strategy of ATOMIC was to provide a view of the atmospheric and oceanic conditions upwind of
the EUREC$^4$A study region. Operations of the *RV Ronald H. Brown* were coordinated with the Wave Gliders and
SWIFTs deployed from the ship, the P-3 aircraft, Saildrone 1064, and BCO and BACO. An additional logistical
objective included recovering the NTAS-17 mooring and replacing it with the NTAS-18 mooring. A third objective
was to triangulate and download data from a Meridional Overturning Variability Experiment (MOVE) subsurface
mooring and related Pressure Inverted Echo Sounders (PIES). MOVE is designed to monitor the integrated deep
meridional flow in the tropical North Atlantic.
Optimal aerosol and flux measurements were made when the ship was pointed into the wind to avoid contamination
by the ship's stack and air flow distortion. Coordinating with the P-3 and Saildrone and deploying the NTAS
Mooring, Wave Gliders, and SWIFTs had the advantage of providing redundant and complimentary data streams but
the disadvantage of requiring the ship to transit away from the wind for maneuvers. In addition, ship transits to
Bridgetown, Barbados for a scheduled in port (Jan. 26 to 28) and a medical emergency (Feb. 3 to 6) were downwind
relative to prevailing northeast trade winds. Periods of unfavorable winds for atmospheric sampling were identified
by relative winds from behind the ship's beam (~ - 90° through 180° to + 90° relative to the bow at 0°). A time
series of relative winds and corresponding high particle number concentrations due to emissions from the ship's
stack (~> 1000 cm$^{-3}$) is shown in Figure 2. These periods have been removed from the aerosol data. Unfavorable
sampling conditions were experienced 15% of the time the ship was at sea at the dates and times indicated in Table
1. Seawater measurements were less accurate when the ship's speed over water was near zero due to mechanical
stirring of the water surface by the ship's propulsion system.
A general timeline of events for Legs 1 and 2 is provided in Sect. 2.1. and 2.2. Descriptions of the instrumentation
onboard the ship and deployed assets are provided in Sect. 2.3 to 2.7 and on the Saildrone, RAAVEN UAS, and
SVPS drifters in Sect. 2.8 to 2.10.

**Figure 2.** Time series of relative wind direction (apparent wind relative to the bow of the ship, negative values are port and positive values are starboard) and particle number concentration ($D_{gn} > 13$ nm) measured on the *RV Ronald H. Brown* during ATOMIC.

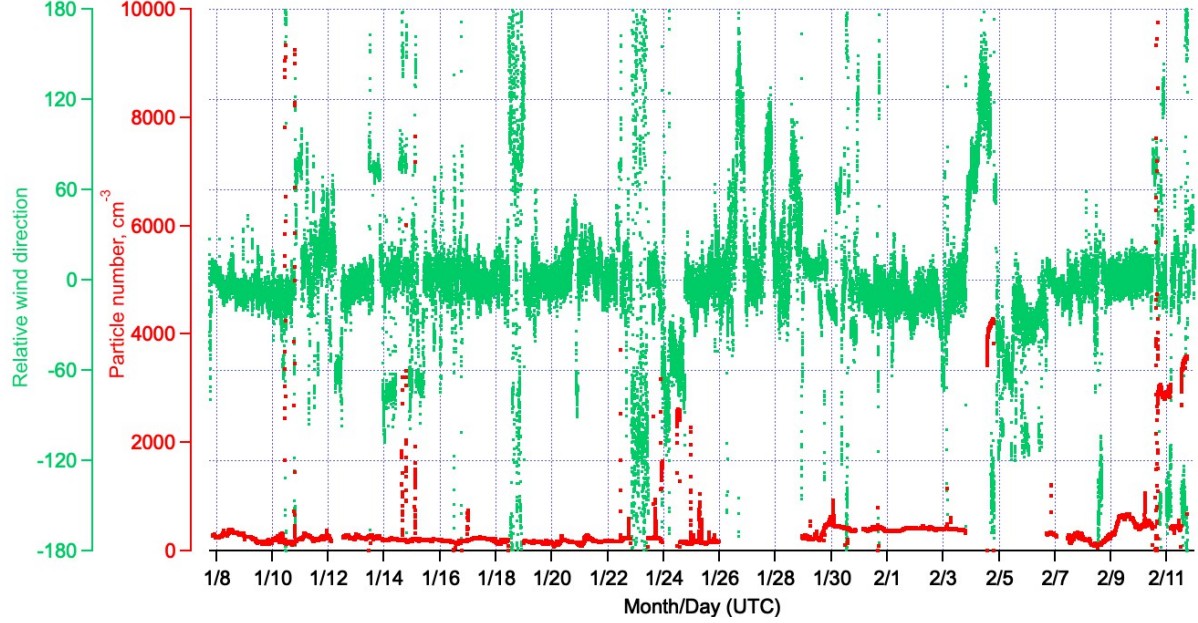

## 2.1. Sampling events during Leg 1

Timeline of events for Legs 1 and 2 are shown in Table 1. Dates and positions of deployment and recovery of assets are listed in Table 2. Times when platforms were within relatively close proximity providing the potential for inter-platform comparisons are given in Table 3. All times reported throughout the paper are in UTC.

During Leg 1, the NTAS mooring was swapped out, Wave Gliders were deployed for the duration of the experiment, and the SWIFTs were deployed and then recovered at the end of the leg. In addition to these logistical operations, measurements were made throughout the leg to characterize atmospheric and oceanic conditions upwind of the EUREC⁴A study region.

The ship departed Bridgetown, Barbados on Jan. 7, 2020 in transit to the NTAS-18 mooring target location at 14°44'N and 50°56'W. Radiosonde launches every 4 hours and continuous atmosphere and sea surface sampling began early on Jan. 8. The latitude and longitude of the four stations occupied during the cruise are listed in Table 1 and shown in Figure 1. Station 1 (S1) was located in the NTAS region. Two Wave Gliders were deployed on Jan. 9 *en route* to S1. Once at S1, early on Jan. 10 a comparison between shipboard and NTAS-17 atmosphere and ocean measurements was conducted. The NTAS-18 mooring was deployed later on Jan. 10. After deployment, the ship transited 55 NM to the northwest of S1 to the MOVE region near 15°27'N and 51°32'W (Figure 1). Unsuccessful attempts were made over a 24 hr period to triangulate the position of the MOVE1-13 mooring and PIES198 and 238.

 The ship left the MOVE region on Jan. 12 at 05:30 to transit back to S1. The MOVE work did not compromise

continuous atmospheric and surface ocean sampling and is not discussed further.

**Table 1.** Timeline of sampling events onboard the *RV Ronald H. Brown* (RHB) including coordination with other
platforms, NTAS operations, downwind transits, and periods on Station. The different colors shown under RHB
correspond to the status of the ship. *NTAS-18 deployed. **NTAS-17 recovered.

| | |
|---|---|
| | Sampling days |
| | Downwind transit |
| | In port |
| | NTAS Operations |
| | MOVE Operations |
| C-BCO | Comparison with BCO |
| C-P3 (Research Flight number) | Coordination with P3 |
| C-SD | Comparison with SailDrone 1064 |
| C-S | Comparison with SWIFTs |
| C-N17 | Comparison with NTAS 17 |
| C-N18 | Comparison with NTAS 18 |
| C-RUAS | Comparison with RAAVEN UAS |
| S1 | Station 1 14°44' N, 50°56'W (NTAS area) |
| S2 | Station 2 14º21'44"N, 53ºW |
| S3 | Station 3 13º54'N, 54º30'W |
| S4 | Station 4 13º51'N, 54º51'36"W |
| Station BCO | Station BCO 13º8'55.7", 59º4'59.2"W |


| | RHB | SWIFTs | WG 245 | WG 247 | CTD | uCTD | NTAS/MOVE |
|---|---|---|---|---|---|---|---|
| Jan 7 | | | | | | | |
| 8 | | | | | | | |
| 9 | S1 | | | | | | |
| 10 | S1, C-N17 | | | | | | N18* |
| 11 | | | | | | | MOVE |
| 12 | S1, C-N18 | | | | | | |
| 13 | S1, C-N18, C-S | | | | | | |
| 14 | C-S | | | | | | |
| 15 | S1, C-N17 | | | | | | |
| 16 | S1, C-N17 | | | | | | N17** |
| 17 | S1, C-P3(RF1) | | | | | | |
| 18 | C-P3 (RF2) | | | | | | |
| 19 | S2 | | | | | | |
| 20 | S2 | | | | | | |
| 21 | S2 | | | | | | |
| 22 | C-S | | | | | | |
| 23 | C-P3(RF3) | | | | | | |
| 24 | C-BCO, C-RUAS | | | | | | |
| 25 | C-BCO, C-RUAS | | | | | | |
| 26 | | | | | | | |
| 27 | | | | | | | |
| 28 | | | | | | | |
| 29 | | | | | | | |
| 30 | C-S, S3 | | | | | | |
| 31 | S3, C-P3(RF5) | | | | | | |
| Feb 1 | S3 | | | | | | |
| 2 | S3 | | | | | | |
| 3 | S3, C-P3(RF6) | | | | | | |
| 4 | | | | | | | |
| 5 | | | | | | | |
| 6 | | | | | | | |
| 7 | | | | | | | |
| 8 | S4, C-SD, C-P3(RF9) | | | | | | |
| 9 | S4, C-SD | | | | | | |
| 10 | S4, C-SD, C-P3(RF10) | | | | | | |

| 11 | C-S, C-WG, C-P3(RF11) | | | | | |
|----|----|----|----|----|----|----|
| 12 | | | | | | |
| 13 | | | | | | |


**Table 2.** Dates (UTC) and positions of deployment and recovery of NTAS moorings, two Wave Gliders, and six
SWIFTs. Assets are listed in order of start and stop times of the data stream. Distance travelled is given for the
SWIFTs and Wave Gliders.


| | Deployment | | Recovery | | |
|----|----|----|----|----|----|
| **Asset** | **Date** | **Position** | **Date** | **Position** | **Distance (NM)** |
| | | | | | |
| | | | **LEG 1** | | |
| Wave Glider 245 | 1/9/20 20:55 | 14° 35' 25" N, 51° 41' 56" W | | | |
| Wave Glider 247 | 1/9/20 20:55 | 14° 35' 13" N, 51° 42' 21" W | | | |
| NTAS-18 | 1/10/20 17:45 | 14° 44' N, 50° 56'W | | | |
| SWIFT 22 | 1/14/20 01:13 | 15° 41' 21" N, 51° 22' 5" W | 1/22/20 19:14 | 15° 3' 29" N, 51° 59' 50" W | 52 |
| SWIFT 23 | 1/14/20 05:11 | 15° 29' 59" N, 51° 18' 54" W | 1/22/20 15:11 | 14° 57' 47" N, 52° 31' 21" W | 62 |
| SWIFT 24 | 1/14/20 07:11 | 15° 24' 42" N, 51° 17' 19" W | 1/22/20 12:13 | 14° 53' 12" N, 52° 49' 5" W | 94 |
| SWIFT 16 | 1/14/20 9:11 | 15° 20' 3" N, 51° 15' 37" | 1/22/20 14:13 | 15° 1' 9", 52° 38' 8"W | 82 |
| SWIFT 25 | 1/14/20 10:12 | 15° 15' 7" N, 51° 13' 46" W | 1/22/20 11:13 | 14° 53' 26"N, 53° 1' 34" W | 60 |
| SWIFT 17 | 1/14/20 18:11 | 15° 34' 31" N, 51° 26' 16" W | 1/22/20 17:14 | 15° 1' 55" N, 52° 18' 56" W | 60 |
| NTAS-17 | | | 1/16/20 10:41 | 14° 49' 28" N, 51° 00' W | |
| | | | | | |
| | | | **LEG 2** | | |
| Wave Glider 245 | | | 2/7 19:55 | 14° 4' 55" N, 54° 17' 12" W | 153 |
| SWIFT 22 | 1/30/20 17:12 | 14° 13' 25" N, 54° 43' 53" W | 2/10/20 17:12 | 14° 20' 35" N, 55° 15' 5" W | 31 |
| SWIFT 16 | 1/30/20 18:13 | 14° 8' 23" N, 54° 40' 51" W | 2/10/20 17:13 | 14° 20' 28" N, 55° 15' 19" W | 36 |
| SWIFT 23 | 1/30/20 19:12 | 14° 3' 31" N, 54° 37' 30" W | 2/10/20 20:15 | 14° 16' 30" N, 55° 25' 14" W | 48 |
| SWIFT 24 | 1/30/20 20:13 | 13° 58' 39" N, 54° 33' 52" W | 2/10/20 23:12 | 14° 20' 19" N, 55° 39' 56" W | 68 |
| Wave Glider 247 | | | 2/11/20 10:54 | 14° 8' 11" N, 55° 57' 9" W | 248 |
| SWIFT 17 | 1/30/20 21:10 | 13° 53' 40" N, 54° 30 '31" W | 2/11/20 15:11 | 13° 55' 47" N, 56° 27' 6" W | 127 |
| SWIFT 25 | 1/30/20 22:13 | 13° 48' 50" N, 54° 27' 6" W | 2/11/20 17:14 | 13° 52' 37" N, 56° 41' 0.6" W | 130 |


**Table 3.** Times when platforms were within relatively close proximity providing the potential for inter-platform
comparisons. Also given are distances between platforms during the comparisons. Results from inter-platform
comparisons reported here are indicated in bold. Distances between RHB and NTAS refer to distance to the mooring
anchor. Distance to buoys were between 0.25 to 3 NM.


| **Platforms** | **Start UTC** | **Stop UTC** | **Distance (NM)** | **Comments** |
|----|----|----|----|----|
| RHB, NTAS-17 | 1/10/20 00:58 | 1/10/20 08:57 | 2.5 (mooring anchor) | Station 1 |
| **RHB, NTAS-18** | **1/12/20 11:30** | **1/13/20 14:00** | **2.9 (mooring anchor)** | **1/12/20 14:06, 19:04; 1/13/20 00:00 CTD casts to 250 m Station 1** |
| **RHB, NTAS-17** | **1/15/20 10:00** | **1/16/20 09:05** | **2.9 (mooring anchor)** | **1/15/20 20:16 CTD cast to 5000 m Station 1** |
| RHB, P-3 | 1/17/20 14:20 | | Within dropsonde circle | P-3 RF1, 7.3 – 7.7 km altitude Station 1 |
| RHB, P-3 | 1/19/20 14:57 | | Within dropsonde circle | P-3 RF2 7.6 km altitude Station 2 |
| RHB, P-3 | 1/23/20 14:06, 19:46 | | Within dropsonde circle | P-3 RF3, 3.2 km altitude 14° 22' 59"N and 55°W Overfly of ship at 150 m at 15:42 |
| RHB, P-3 | 1/31/20 16:25 | | Within dropsonde circle | P-3 RF5, 7.4 km altitude Station 3 |
| RHB, P-3 | 2/3/20 14:13 | | Within dropsonde circle | P-3 RF6, 7.7 km altitude Station 3 |
| RHB, P-3 | 2/9/20 05:57 | | Within dropsonde circle | P-3 RF9, 7.5 km altitude Station 4 |
| RHB, P-3 | 2/10/20 05:46 | | Within dropsonde circle | P-3 RF10, 7.5 km altitude Station 4 |
| RHB, P-3 | 2/11/20 | | | P-3 RF11, 7.5 km altitude |

| | 10:26 | | | | Station 4 |
|---|---|---|---|---|---|
| **RHB, SD 1064** | **2/8/20 9:30** | **2/10/20 18:50** | **0.7 to 3.6** | | **2/8/20 9:30 – 18:10 SD was 2.8 – 3.6 NM upwind, 2/8 19:00 0 2/10 18:50 SD was 0.7 – 0.8 NM from ship Station 4** |
| **RHB, BCO** | **1/24/20 18:20** | **1/25/20 23:40** | **20** | | **RHB located directly upwind of BCO** |


A comparison of atmospheric and oceanic parameters measured onboard the ship and NTAS-18 was conducted Jan.
12 to 13. The comparison included a CTD (conductivity, temperature, and depth) sensor mounted on the ship's
rosette and conductivity and temperature sensors attached to the NTAS mooring line. While waiting for the weather
to calm down enough to recover NTAS-17, 6 SWIFTs were deployed. The ship transited 55 NM to the northwest
and deployed the first SWIFT (22) on Jan. 14 at 01:13 UTC at 15° 41' 21" N, 51° 22' 5" W (Fig. 1, Table 2).
Following a southeast track, the remaining 5 SWIFTs were deployed 5 to 12 NM apart across horizontal gradients in
ocean surface current and temperature. SWIFT17, the second one deployed, was recovered due to the failure of a 3-
D sonic anemometer. The ship returned to SWIFT17 to swap out the anemometer. It was re-deployed near its
original position on Jan. 14 at 18:11 UTC. After each SWIFT deployment, underway CTD (uCTD) casts were
performed to a depth of 50 m for comparison to SST and salinity measured onboard the SWIFTs and to understand
the ocean mixed layer structure at the beginning of each SWIFT Lagrangian drift. In addition, the ship sat near each
SWIFT for at least an hour after deployment for a comparison of measured near surface atmospheric and surface sea
water parameters.

The ship returned to S1 and conducted a second comparison with NTAS-17 on Jan. 15 to 16, including a CTD cast
with the ship's rosette and sensors on the NTAS mooring line. NTAS-17 was recovered on Jan. 16. The ship stayed
at S1 and was within the P-3's dropsonde circle during its first flight (Research Flight 1 or RF1) on Jan. 17 from
15:30 to 16:40. A first comparison between the uCTD and the CTD on the ship's rosette for temperature and salinity
was conducted on Jan. 17 at 22:36. The ship's CTD cast went to a depth of 500 m.

With the NTAS and MOVE work finished, the ship transited downwind on Jan. 18 for 14.5 hours to Station 2 (S2)
located at 14º21'44"N and 53ºW (Fig. 1a). This location was downwind of the projected paths of the SWIFTs but
still upwind of the EUREC⁴A study region. During the transit, aerosol and flux measurements were compromised by
relative winds abaft the beam but surface ocean and meteorological measurements as well as radiosonde launches
continued. In addition, uCTD casts to 100 m depth were made every hour to investigate a large-scale SST gradient
between NTAS and S2. The ship briefly slowed to 2 to 4 kts for each cast.

The ship reached S2 on Jan. 19 at 01:30 UTC and turned into the wind for optimal aerosol and flux measurements.
Underway CTDs were conducted to a depth of 100 m every 6 hr. The second overflight of the P-3 (RF2) occurred
on Jan. 19 at 14:57 UTC with the *RV Ronald H. Brown* within the aircraft's dropsonde circle. A second comparison
between the uCTD and the CTD on the ship's rosette was conducted on Jan. 21 at 16:15 with the ship's CTD
reaching a depth of 150 m.

On Jan. 22 at 07:30 UTC, the ship left S2 to recover the SWIFTs before the end of Leg 1. The SWIFTs had drifted
between 53 and 103 NM to the southwest with those deployed at the more southern locations drifting the furthest
(Fig. 1a, Table 2). The ship transited 32 NM to the north to reach the southernmost SWIFT and then followed a
course to the northeast recovering the remaining SWIFTs which were 7 to 24 NM apart. Once all SWIFTs were
onboard (Jan. 22 19:14), the ship transited 180 NM to the southwest to 14° 22' 59”N and 55°W to be in the center of
the P-3's dropsonde circle the next day. Aerosol and flux measurements were compromised during the transit due to
the relative wind being abaft the beam.
The ship reached the designated position on Jan. 23 at 10:30, turned into the wind for optimal aerosol and flux
measurements, and was within the P-3's dropsonde circle on Jan. 23 at 14:06 (RF3). Later in the flight (15:42), the
P-3 flew over the ship at an altitude of 150 m. This flyby was the closest the P-3 was to the ship during the
ATOMIC campaign while all instrumentation was operational. At 22:00 the ship started the 250 NM transit back to
Bridgetown with a planned stop upwind of BCO/BACO for a measurement comparison. Initially, relative winds
were from the port side of the ship at -100° relative to the bow but 6 hrs into the transit they shifted to a relative
direction of -50° due to a change in true wind direction and the ship's course, making for better conditions for
aerosol and flux measurements. Radiosonde launches were halted on Jan. 24 at 2:45 near 56°W with the knowledge
that sondes launched from the *RV Meteor* and BCO could be used to fill in the gap. The ship arrived at the
comparison point 20 NM east of BCO (13° 8' 55.7"N, 59° 4' 59.2"W) at 18:20 on Jan. 24 and stayed until Jan. 25 at
23:40 (Fig. 1a). Underway CTDs were conducted approximately every 2 hours until Jan. 25 at 21:58.
The ship ended Leg 1 with a transit around the southern end of Barbados and into Bridgetown with an arrival on Jan.
26 at 12:15 for open house and outreach activities to be conducted on Jan. 27.
**2.2. Sampling events during Leg 2**
During Leg 2, the SWIFTs were deployed at the beginning of the leg and then recovered along with the Wave
Gliders at the end of the leg. Similar to Leg 1, measurements were made throughout the leg to characterize
atmospheric and oceanic conditions upwind of the EUREC[4]A study region.
The *RV Ronald H. Brown* left Bridgetown at 22:15 on Jan. 28 and headed for Station 3 (S3) located 290 NM to the
northeast of BCO/BACO at 13° 54' 0"N and 54° 30' 0"W (Fig. 1b). S3 was roughly halfway between BCO/BACO
and NTAS. Radiosonde launches began on Jan. 29 at 6:45 and continued every 4 hrs. The ship veered off its NE
track on Jan. 29 at 20:18 and turned to the southeast to map the spatial orientation of SST fronts with gradients
around 0.75°C for determining where to deploy SWIFTs. When done with mapping, the ship went north on Jan. 30
at 4:15 arriving in the vicinity of S3 and Wave Glider 245 at 08:00. Wave Glider 245 was recovered to replace
malfunctioning sensors.

The ship zigzagged to the northwest and then northeast until reaching 14° 13' 25" N and 54° 43' 53" W on Jan. 30 at
17:12 where the first SWIFT deployment of Leg 2 took place (Fig. 1d, Table 2). The remaining SWIFTs were
deployed on a southeast track approximately 6 NM apart. After each SWIFT deployment, uCTD casts were
performed to a depth of 100 m to provide a subsurface context for SWIFT measurements. During each cast, the ship
moved into the wind at 0.5 kts. Wave Glider 245 was re-deployed on Jan. 30 at 18:08 after the last SWIFT was put
in the water. The ship then transited back to S3, arriving 5 hrs later at 23:09. During this 6 hr period, as the ship was
maneuvering to deploy SWIFTs, relative winds were from the port side between -50 to -100 degrees compromising
aerosol and flux measurements.

The ship remained at S3 until Feb. 3 at 15:00 to characterize diurnal variations in oceanic and atmospheric
conditions and to be in position for the P-3's RF5 and RF6. Continuous atmospheric and surface ocean
measurements were made, radiosondes were launched every 4 hrs, and uCTD casts were conducted every 2 hrs.
Four comparisons between the uCTD and the CTD on the ship's rosette were conducted between Feb. 1 and Feb. 3
with the ship's CTD reaching a depth of 400 m. The ship was at the center of the P-3's dropsonde circle on Jan. 31
at 16:25 (RF5) and Feb. 3 at 14:13 (RF6).

On Feb. 3 at 19:30 the ship headed back to Bridgetown for a medical emergency. Aerosol and flux measurements
were compromised due to relative winds abaft the beam. Radiosonde launches continued every 4 hours. The last
launch before reaching port was on Feb. 4 at 10:45. The ship arrived in Bridgetown on Feb. 4 at 19:00.

The ship departed Bridgetown on Feb. 6 at 16:00 and headed northeast to Station 4 (S4) located at 13º51'N and
54º51'36"W, 21.2 NM southwest of S3. Atmospheric measurements resumed along with radiosonde launches every
4 hrs. The ship arrived at S4 on Feb. 8 at 01:00 but left 6 hrs later to recover Wave Glider 245 because it was
experiencing navigation problems that could have endangered the vehicle. The Wave Glider was recovered 36 NM
to the northeast of S4 (14° 4' 55" N, 54° 17' 12" W) on Feb. 8 at 12:45. Aerosol and flux measurements were
compromised during the downwind transit back to S4 between 12:45 and 16:25. Once back on station, optimal
aerosol and flux measurements resumed along with uCTD casts every 2 hrs. A CTD cast to a depth of 1000 m with
the ship's rosette was conducted on Feb. 8 at 17:00 for comparison to the uCTD.

Still at S4, the ship was within the P-3's night time dropsonde circle on Feb. 9 (RF9) at 5:57. The NOAA PMEL-
operated Saildrone 1064 completed a first leg between BCO and NTAS and then sailed near the ship for a
comparison of fluxes and measured meteorological and seawater parameters. The Saildrone was 2.8 to 3.6 NM
upwind of the ship between Feb. 8 from 9:30 to 18:10 and within 0.7 to 0.8 NM of the ship between Feb. 8 19:00
and Feb. 10 18:50. Two final comparisons between the uCTD and the CTD on the ship's rosette were conducted on
Feb. 8 and Feb. 9 with the ship's CTD going to depths of 1000 and 400 m, respectively. The ship remained at S4 for
the P-3's second night flight (RF10) and was within the dropsonde circle on Feb. 10 from 05:46 to 06:42. The ship's
final coordination with the P-3 occurred during a combination research and sightseeing flight with press (RF11) on
Feb. 11. The ship was not within the dropsonde circle but was flown over at sunrise at 10:26.

The ship remained at S4 until Feb. 10 at 12:00 at which point aerosol measurements were ended and the ship began
the transit to recover SWIFTs and Wave Glider 247. Recovery operations were conducted between Feb. 10 15:00
and Feb. 11 18:15. The four SWIFTs (16, 22, 23, and 24) that were initially deployed to the north between 14° 13'
25" and 13° 58' 39" N drifted to the northwest travelling a total distance ranging from 31 to 68 NM (Table 2, Fig.
1b). The two SWIFTs (17 and 25) deployed to the south between 13° 53' 40" N and 13° 48' 50" N initially drifted to
the southwest, each traveling 130 NM. The ship transited to the northeast to pick up the northern cluster of SWIFTs
first, staying near each asset for up to 1.5 hrs for a comparison of measured atmospheric and oceanic parameters.
The ship then did several back-and-forth tracks between the position of Wave Glider 247 and SWIFT 17 mapping a
SST front before recovering the Wave Glider and the last two SWIFTs.

After the SWIFTs and Wave Glider were recovered, the ship started a northeast transit on Feb. 11 around 19:30
across a SST front in the upwind direction to study air-sea interaction and atmospheric and oceanic mixed layer
variability. Underway CTDs were made continuously. On Feb. 12 at 06:00, the ship began the southwest transit back
to Bridgetown for the final time. Atmospheric sampling was compromised during the downwind transit. The last
radiosonde launch occurred on Feb. 12 at 10:45. The ship arrived in port on Feb. 13 at 10:00.

**2.3. NTAS operations and measurements**

NTAS was established to provide accurate air-sea flux estimates and upper ocean measurements in a region with
strong SST anomalies and the likelihood of significant local air-sea interaction on interannual to decadal timescales
(Weller, 2018;Bigorre and Galbraith, 2018). The station is maintained at a site near 15°N and 51°W through
successive mooring turnarounds. During Leg 1, the Upper Ocean Processes Group of the Woods Hole
Oceanographic Institution (WHOI) and crew of the *RV Ronald H. Brown* deployed the NTAS-18 mooring and
recovered the NTAS-17 mooring at nearby sites. Both moorings used Surlyn foam buoys as the surface element.
These buoys are outfitted with two Air–Sea Interaction Meteorology (ASIMET) systems (Colbo and Weller, 2009).
The ASIMET system measures, records, and transmits via Iridium satellites the surface meteorological variables
necessary to compute air–sea fluxes of heat, moisture and momentum. The upper 160 m of the mooring line are
outfitted with oceanographic sensors for the measurement of temperature, salinity and velocity. Information on the
instruments providing real-time data, measured atmospheric and oceanic parameters, and height/depth of the
measurements on the NTAS mooring are provided in Table 4.


**Table 4.** Instrumentation providing real-time data onboard the NTAS mooring.

| Instrument | Measured/derived quantities, raw sampling interval | |
|---|---|---|
| | | |
| | *Atmospheric parameters* | **Height (m)** |
| ASIMET system | Bulk air-sea fluxes, relative humidity, temperature, pressure, wind speed and direction, precipitation rate, longwave radiation, shortwave radiation, 1 min | 3 |
| | | |
| | *Oceanic parameters* | **Depth (m)** |
| ASIMET system | Sea surface temperature and salinity, 1 min | 0.8 |
| Seabird (SBE-37 IM) | Temperature and salinity, 5min | 10 |
| NORTEK Aquadopp | Currents, 20 min | 13 |
| Seabird (SBE-37 IM) | Temperature and salinity, 5min | 25 |
| Seabird (SBE-37 IM) | Temperature and salinity, 5min | 40 |
| Seabird (SBE-37 IM) | Temperature and salinity, 5min | 55 |
| Seabird (SBE-37 IM) | Temperature and salinity, 5min | 70 |

ASIMET data are sampled and recorded internally every minute. The oceanographic measurements are recorded either every 5 min or 10 min for temperature and salinity (depending on the instrument type) and 20 min or 1 hr for currents. The NTAS-18 mooring was deployed on Jan. 10 at 14° 44' N, 50° 56' W with anchor drop at 17:45 in 5055 m of water. The NTAS-17 mooring was recovered on Jan. 16 with anchor release at 10:41. Both buoys have a watch circle of about 2 NM from their respective anchors and were separated by about 6 NM during the January 10 to 16 period allowing for comparisons of measured ocean and atmosphere parameters. Atmospheric data from NTAS-17 and NTAS-18 were combined for comparison to measurements onboard the *RV Ronald H. Brown* (Sect. 4.2.1).Wind speed, air temperature, and specific humidity were adjusted to a height of 10 m and neutral atmospheric stability using the COARE 3.6 bulk model for the comparison (Fairall et al., 2003;Edson et al., 2013). NTAS data in the ATOMIC archive only include data collected during the ATOMIC campaign.

On Apr. 8, 2020 at 08:00 UTC, the NTAS-18 buoy went adrift. It meandered slowly toward the Caribbean for 7 months until being recovered on Oct. 20, 2020. NTAS-19 was deployed on Oct. 22, 2020.

**2.4. Shipboard atmospheric measurements**

Instrumentation onboard the *RV Ronald H. Brown* for the measurement of atmospheric and aerosol parameters is listed in Table 5. Locations of instruments on deck are shown in Figure 3. NOAA's Physical Science Laboratory (PSL) collected data to enable a deeper understanding and quantification of cloud processes, the environments in which they either grow or dissipate, how the ocean and atmosphere interact, and the spatial variability of these processes. Instrumentation mounted on the bow mast and forward O2 deck (two levels above the main deck) measured sea-surface meteorological properties, rain rate, radiative fluxes, and air-sea turbulent fluxes using bulk, eddy covariance, and inertial dissipation methods (Fairall et al., 1997;Fairall et al., 1996;Fairall et al., 2003;Edson et al., 2013). Vertical profiles of backscatter from a ceilometer mounted on the forward O3 deck (three levels above the main deck) provided cloud base height and temporal cloud fraction. For comparison with other platforms (NTAS and Saildrone 1064), wind speed, air temperature, air pressure, and specific humidity were adjusted to a height of 10

m using the COARE 3.6 bulk algorithm. Final data products of meteorological and navigation data are 1-min and
10-min averages of high-resolution raw data (see Table 5 for raw sampling intervals). The data are time-stamped at
the beginning of the 1- and 10-min period. Fluxes were calculated at 10-min resolution, then interpolated to 1-min.

University of Miami (UM) provided high resolution measurements of cloud and rain to better understand the
relationship between cloud properties and cloud spatial organization as a function of cloud mesoscale organization,
in particular rain and the associated atmospheric cold pools (Stevens et al., 2020;Löffler-Mang and Joss, 2000). Two
collocated Parsivel disdrometers mounted on the forward O3 deck provided precipitation intensity, drop number,
and equivalent radar reflectivity. A sky camera provided a 50° field of view oriented horizontally off the starboard
side of the ship every 4 sec. A microwave radiometer was deployed to provide cloud liquid water path estimates but
its data acquisition was unsuccessful and no data are available. A Marine Atmospheric Emitted Radiance
Interferometer (M-AERI) was mounted on the port side O2 deck rail (2 levels above the main deck) (Minnett et al.,
2001). It measured the spectra of infrared emission from the sea surface and atmosphere for the derivation of skin
**Table 5**. Instrumentation onboard the *RV Ronald H. Brown* for the measurement of atmospheric and aerosol
parameters. The O2 and O3 decks were two and three levels above the main deck, respectively. [a]Aerosol inlet was
located on the O2 deck, 18 m.a.s.l. Final data products of meteorological and navigation data are 1-min and 10-min
averages of high-resolution raw data and time-stamped at the beginning of the 1- and 10-min period. Fluxes were
calculated at 10-min resolution, then interpolated to 1-min for those files.

| Instrument | Measured/derived quantities, raw sampling interval | Location |
|---|---|---|
| *Atmospheric parameters* | | |
| Gill WindMaster Pro 3-axis ultrasonic anemometer | Wind vector, stress, and sensible heat flux, 0.1 sec | Bow mast |
| Optical precipitation sensor, OSI Inc., ORG-815 DA | Rain rate, 5 sec sampling, collected/recorded every 1 min | Bow mast |
| Li-COR 7500 Gas Analyzer | Water vapor density, turbulent latent heat flux, 0.1 sec | Bow mast |
| Vaisala HMT335 | Air temperature, humidity, 1 min | Bow mast |
| Vaisala PTB220 | Atmospheric pressure, 1 min | O2 deck |
| 2 Eppley PSPs (Pyranometer) | Shortwave radiation, 1 min | O2 deck |
| 2 Eppley PIRs (Pyrgeometer) | Longwave radiation, 1 min | O2 deck |
| Systron and Donner MP-1 6-axis motion detector system | 3-D ship acceleration, 0.1 sec | Bow mast |
| Vaisala CL31 Ceilometer | Vertical profiles of backscatter from refractive index gradients, cloud base height, cloud fraction, 15 sec sampling from 0-7.7 km with 10 m vertical spacing | O3 deck |
| 2 Parsivel optical rain gauges, 650 and 780 nm | Rain rate, equivalent radar reflectivity, particle number | O3 deck |
| StarDot Camera, NetCam XL | Pointed to starboard, field of view of 50°, image captured every 4 sec | O3 deck |
| Doppler lidar λ=1.5 μm | Atmospheric vertical velocity and backscatter intensity, horizontal wind profiles, estimates of cloud base and mixed layer heights; 0.5 sec | O2 deck |
| W-band (95.56 GHz) Doppler vertically pointing cloud radar | Vertical profiles of non-precipitating and lightly-precipitating clouds from 100 m to 4.2 km with 30 m vertical resolution every 0.5 sec | O2 deck |
| Dual-flow, two filtered radon detector | $^{222}$Rn, 30 min | O3 deck |
| Vaisala WXT536 | T, RH, rain rate; 1 sec | O2 deck |
| Picarro water vapor isotope analyzer (L2130-fi) | Water vapor concentration and isotopic composition, 0.2 sec | Aerosol inlet[a] |
| Vaisala RS-41 radiosondes | Profiles of T, RH, P, and winds every 4 hrs | Main deck |
| Thermo Environmental Model 49C | Ozone, 1 sec | Inlet at 18 m.a.s.l. |
| *Aerosol Properties* | | |
| Collection with multi-jet cascade impactors and analysis by ion chromatography, thermal-optical, gravimetric, and XRF analysis | Size segregated concentrations of Cl$^-$, NO$_3^-$, SO$_4^=$, methanesulfonate (MSA$^-$), Na$^+$, NH$_4^+$, K$^+$, Mg$^{+2}$, Ca$^{+2}$, organic carbon, elemental carbon, trace elements; hours | Aerosol inlet[a] |
| DMPS and TSI 3321APS | Number size distribution 0.02 to 10 μm, 5 min | Aerosol inlet[a] |

| TSI 3025A, 3760A, 3010 | Number concentration > 3, 13, 13 nm; 1 sec | Aerosol inlet[a] |
|---|---|---|
| TSI 3563 Nephelometer | Sub-1.1 and sub-10 μm light scattering and backscattering; 450, 550, 700 nm; 60% RH; 1 sec | Aerosol inlet[a] |
| TSI 3563 Nephelometers | Sub-1.1 μm scattering f(RH); 450, 550, 700 nm; dry and 80%RH; 1 sec | Aerosol inlet[a] |
| Radiance Research PSAP | Sub-1.1 and sub-10 μm light absorption; 467, 530, 660 nm; dry | Aerosol inlet[a] |
| DMT CCNC | Sub-1.1 μm cloud condensation nuclei concentration, 0.1 to 0.6% S, 1 sec | Aerosol inlet[a] |
| Solar Light Microtops Sunphotometer | Aerosol Optical Depth; 380, 440, 500, 675, 870 nm | O3 deck |


**Figure 3.** Instrumentation onboard the *RV Ronald. H. Brown* for the measurement of atmospheric and oceanic parameters located on a) the bow mast and forward O2 deck and b) port side O3 deck. Asset deployments are shown for c) NTAS mooring, d) ship's rosette with CTD and Niskin bottles, e) uCTD, f) SWIFT, g) Wave Glider, and h) radiosonde. Also shown are i) P-3 fly over of the ship on Jan. 23 and j) Saildrone upon its return to the U.S. (Newport, RI) from Barbados. Not shown are disdrometers on the port O3 deck and camera on the starboard O3 deck.

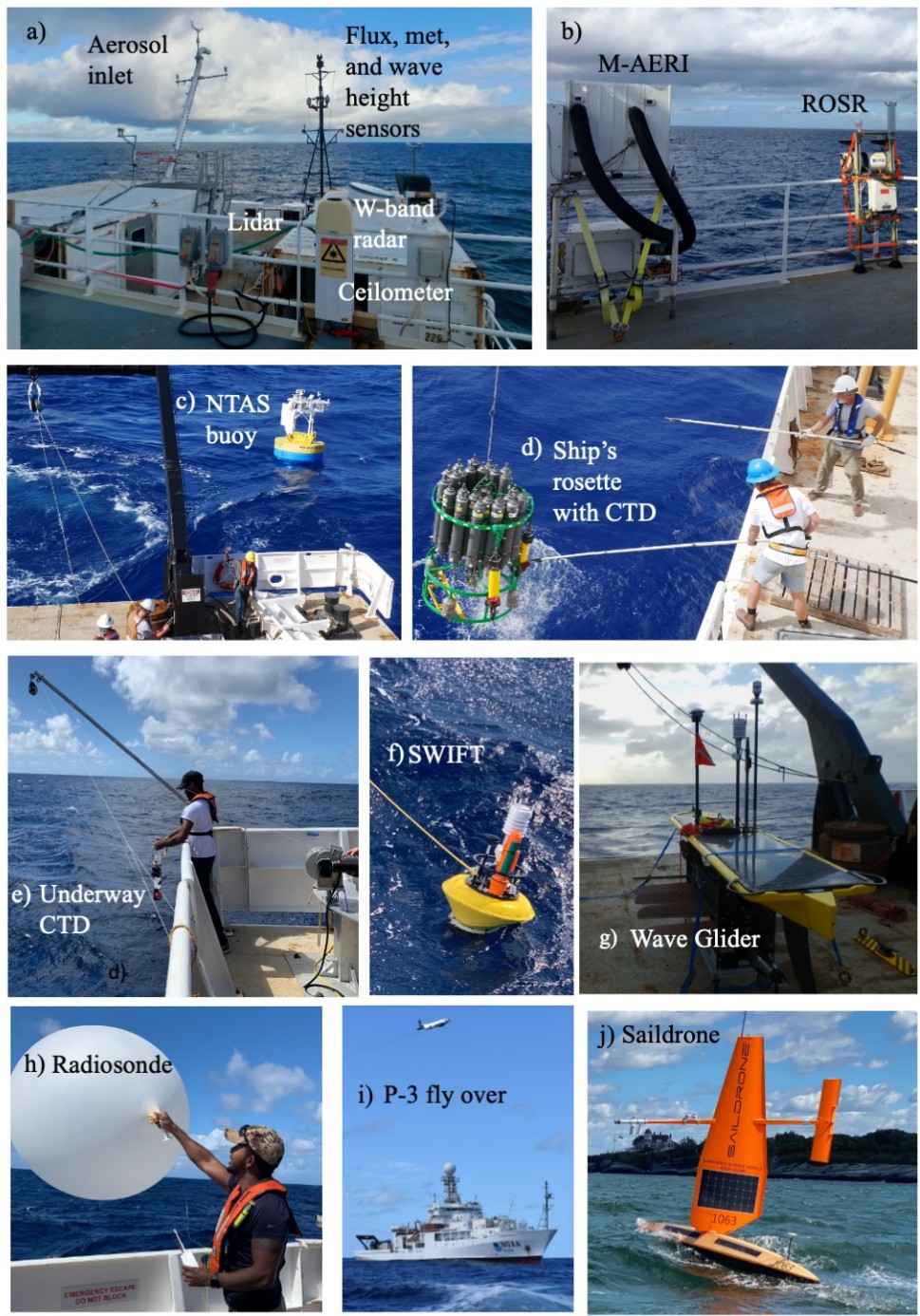

sea surface temperature and lower troposphere profiles of temperature and humidity (Szczodrak et al., 2007). A W-
band Doppler vertically pointing cloud radar was housed in a container on the O2 deck for the measurement of
vertical profiles of non-precipitating and lightly-precipitating clouds (Moran et al., 2012). The radar was not
functional during Leg 1 and operated with a 10 dB attenuator on Leg 2 that prevented detection of non-precipitating
clouds. Although the loss of this information limited the ship-based based observations of non-precipitating cloud,
data from the shipboard ceilometer and Doppler radar and the cloud radar on the P-3 will be used to fill in gaps.

NOAA's Chemical Sciences Laboratory (CSL) operated a microjoule class, pulsed Doppler lidar (microDop)
operating at a wavelength of 1.5 μm to assess atmospheric turbulence, aerosol backscatter intensity, and horizontal
winds (Schroeder et al., 2020). The lidar was mounted on the forward O2 deck. The system was motion stabilized
while staring vertically to within 0.25 degrees of zenith. Ship motion projected onto the line-of-site velocity
measurement was estimated and removed using a 6-axis inertial navigation unit (INU). The INU allowed the lidar to
measure the mean and turbulent motions of aerosol in clear air and cloud scatterers with a spatial and temporal
resolution of 33.6m and 2Hz respectively. The first valid gate was 75m above the ocean surface. The maximum
height of valid data depends on the availability of aerosol scattering targets.  Typically, the instrument provided data
through the top of the marine atmospheric boundary layer, in the presence of elevated dust layers to 3 km, and
clouds to a height of 7 km. The lidar pointed vertically 95% of the time to sample updrafts and downdrafts in the
subcloud mixed layer and in the interstitial trade cumulus boundary layer and spent 2 minutes of every hour
performing a 65° elevation, full azimuthal scan to measure horizontal wind profiles. Real-time quicklooks of
backscatter intensity profiles showing strongly scattering cloud base and updraft structures were available for
awareness of the clouds and turbulent mixed layer throughout the cruise. Cloud base height (CBH) was retrieved by
applying Haar wavelet covariance transforms to the backscatter intensity profiles.

Oregon State University and the National Center for Atmospheric Research (NCAR) operated a Picarro water vapor
isotope analyzer on Leg 2 of the cruise to investigate processes that shape the atmosphere's humidity structure and
its variations. The spectroscopic analyzer measured water vapor concentration and its isotopic composition, the
isotope ratios of oxygen ($^{18}O/^{16}O$) and hydrogen (D/H). All three quantities were measured continuously at 5 Hz
frequency via the aerosol inlet on the O2 deck at 18 m above sea level (m.a.s.l.). Complementary gas-phase water
isotopic measurements were made from the P-3, at BCO, from the French ATR aircraft, and aboard German and
French research vessels. Rainwater and seawater were also collected from the ship platforms for future offline
analysis. Surface sea water and water column samples from CTD casts were also collected to investigate the upper
ocean mixing and the freshwater balance to be evaluated in the context of air-sea gas exchange and upper ocean
circulation.

The goals of NOAA's Pacific Marine Environmental Laboratory (PMEL) were to assess the impacts of aerosols on
clouds and direct aerosol light scattering and absorption on the temporal variability of net radiation reaching the
ocean surface and SST for the conditions of a well-mixed boundary layer. Measurements included aerosol chemical
composition, total number concentration, number size distribution, light scattering and its dependence on relative
humidity, light absorption, and cloud nucleating ability. Aerosol instrumentation was housed in two containers on
the O2 deck. All instruments drew sample air from an inlet 18 m.a.s.l. mounted on top of one of the O2 deck vans
(Bates et al., 2002) (Fig. 3). Aerosol optical depth (AOD) was measured using Microtops hand held sunphotometers.
The raw Microtops data were processed by the NASA Maritime Aerosol Network in conjunction with the Aerosol
Robotic Network (Smirnov et al., 2009). In addition, $^{222}$Rn was measured for its use as a tracer of continentally-
influenced air masses (Whittlestone and Zahorowski, 1998) and $O_3$ was measured for its use as an indicator of
entrainment from the upper troposphere.

Radiosondes were launched throughout the ATOMIC campaign to provide information about the temporal evolution
and vertical structure of the boundary layer, upper atmosphere, and clouds. A total of 97 radiosondes (Vaisala RS41-
SGP) were launched from the fantail during Leg 1 and 66 were launched during Leg 2. There were 6 launches per
day at 02:45, 06:45, 10:45, 14:45, 18:45, and 22:45 UTC. Vertical profiles of pressure, temperature, relative
humidity, and winds were measured from the surface to approximately 25 km. Measurements were also made during
the radiosondes' descent. Data were communicated to the Global Telecommunications System (GTS) following
each sounding via email to the U.S. National Weather Service and via FTP to MeteoFrance. The data were put into
10 m altitude bins and merged with the EUREC[4]A sounding network. Raw (Level-0), quality-controlled 1-second
(Level-1), and vertically gridded (Level-2) data in NetCDF format are available to the public at AERIS
(https://doi.org/10.25326/62). The methods of data collection and post-processing can be found in (Stephan et al.,

434   2020).


Radiosonde operations were suspended on the ship west of ~56°W when the ship transited to Bridgetown for the
planned in port (Jan. 24 at 2:45) and an emergency medical evacuation (Feb. 4 at 10:45). Soundings from BCO were
stitched together with those from the ship to allow for an uninterrupted data record over the entire cruise.

The Lifted Condensation Level (LCL) was calculated from the BCO-RHB radiosonde data record and assumed to
represent Cloud Base Height (CBH). The LCL (in m) was calculated as

443          $$LCL = (T_{50} - T_{d,50}) \times 125 + 50 \qquad\qquad\qquad (1)$$


where $T_{50}$ is temperature and $T_{d,50}$ is dew point, both at 50 m height (Espy, 1836; Bolton, 1980). The lowest altitude
considered was 50 m to avoid contamination by the temperature and relative humidity near the ship's deck and to
minimize the effect of vertical gradients in the surface layer. Since the calculation started at 50m, 50 was added to
the LCL.

**2.5. Shipboard oceanic measurements**

Instrumentation onboard the *RV Ronald H. Brown* for the measurement of oceanic parameters is listed in Table 6.
Locations of instruments mounted on the deck are shown in Figure 3. As stated above, UM's M-AERI, located on
the port side forward O2 deck, measured sea surface skin temperature (Minnett et al., 2001).
**Table 6**. Instrumentation onboard the *RV Ronald H. Brown* for the measurement of seawater parameters. O2 deck is
two levels above the main deck.

| Instrument | Measured /derived quantities, raw sampling interval | Location |
|---|---|---|
| Marine Atmospheric Emitted Radiance Interferometer (M-AERI) | Sea surface skin temperature, 5 – 7 min averages | O2 deck |
| Remote Ocean Surface Radiometer (ROSR) | Sea surface skin temperature, 5 min averages | O2 deck |
| Floating YSI 46040 Thermistor (Sea snake) | Sub-skin sea surface temperature, ~ 0.05 m depth, 1 sec | Deployed off port side with outrigger |
| Riegl 1-D laser altimeter | Wave height and period, 10 min averages | Bow mast |
| Seabird 9+ CTD | At station conductivity (salinity), temperature, depth (pressure), PAR, fluorescence, and oxygen | Deployed off starboard side, main deck |
| Seabird SBE45 thermosalinigraph Seabird SBE38 thermistor | Seawater temperature, conductivity (salinity), 1 sec | 5.3 m below the surface |
| Acoustic Doppler Current Profiler 75 kHz (ADCP) | Current velocity across 2 depth ranges depending on mode. Narrowband: 29-892 m. Broadband: 17-333 m. 5 min sampling. | Ship's hull |
| RBR Concerto underway CTD + Tuna Brute winch (uCTD) | Conductivity (salinity), temperature, and depth (pressure) from the surface to 60 or 130 m depending on cast | Deployed off starboard aft quarter |


During Leg 1, the Applied Physics Laboratory at the University of Washington (APL-UW) also measured sea
surface skin temperature with a Remote Ocean Surface Radiometer (ROSR) located near the M-AERI. PSL
measured sub-skin temperature at approximately 0.05 m depth with a floating thermistor (a.k.a sea snake) deployed
off the port side. A skin temperature value was estimated by the COARE algorithm using the sea snake data as input
(Fairall et al., 1996;Fairall et al., 1997). This algorithm accounts for the cool skin present in the upper ~0.2-1 mm
and any potential diurnal warm layers in the upper ~10 m. This COARE-calculated skin T and the current-relative
wind were used to compute bulk, eddy covariance, and inertial dissipation air-sea fluxes (Fairall et al., 1997;Fairall
et al., 2003). The COARE 3.6 algorithm estimated wave parameters using wind as input. The parameterization is
based on fits to the Banner and Morison (2010) wave model and the flux database collected by NOAA PSL,
University of Connecticut, and Woods Hole Oceanographic Institution (Fairall et al., 2003;Edson et al., 2013). PSL
also measured significant wave height and period with a 1-dimensional downward looking RIEGL laser altimeter
mounted on the bow mast.

The ship's rosette-mounted CTD was intermittently deployed off the starboard main deck for comparison to the
uCTD, Wave Gliders, SWIFTs, and NTAS moorings. Water was collected from the Niskin bottles for analysis of the
isotopic composition of oxygen and hydrogen. In addition, the ship had an underway seawater sampling system
consisting of a thermistor SBE38 located at the intake valve on the hull and a thermosalinograph SBE45 located
inside the ship. These sensors produced underway measurements of temperature and conductivity (salinity) from
water sampled at ~5.3 m below the surface. The values recorded may be representative of seawater properties
shallower in depth due to an unknown amount of mixing along the hull of the ship that is dependent on currents,
ship speed, and waves. The ship also had a 75 kHz acoustic Doppler current profiler (ADCP) for the measurement of
currents at depths greater than ~ 17 m.

UW deployed an underway CTD (uCTD) for the measurement of conductivity (salinity), temperature, and pressure
(depth) to assess variability in the upper 60 to 130 m of the water column (Mojica and Gaube, 2020). The uCTD was
deployed off the starboard aft quarter. Initially, the probe was lowered by hand with line pre-measured to 50 m.
Casts were completed more frequently and with an electric winch after the NTAS mooring work was done which
freed up deck space. During Leg 2, a cast with the ship's CTD was conducted every day at 13:00 (Jan. 31, Feb. 1
and 2) or 17:00 (Feb. 3, 8, and 9) shortly after a uCTD cast. These casts were used to correct the uCTD conductivity
data which had a small offset due to interference from the sensor guard. A transect of intensive uCTD data was
collected when the ship transited from NTAS (S1) to S2 on Jan. 18. While at S2, uCTD casts were conducted every
1 to 4 hrs. In addition, uCTD casts were conducted every 2 hrs during the majority of Leg 2 when the ship was
stationary. The frequency of uCTD sampling increased to every 10 min between 13:00 and 15:15 on Feb. 9 to study
heaving of periodic internal waves located at the base of the mixed layer (60-80 m depth) and for 7 hrs at the end of
Leg 2 on Feb. 11 and 12 as the ship transited across a strong SST front in the upwind direction. uCTD casts were
also performed when deploying or recovering the SWIFTs and Wave Gliders for comparison purposes.

**2.6. Wave Glider measurements**

Two Wave Gliders (serial numbers 245 and 247) operated by APL-UW were deployed within 15 minutes of each
other on Jan. 9 (Fig. 1a and Table 2). The Wave Gliders greatly increased the sampling of spatial inhomogeneities in
atmospheric and oceanic properties as well as bulk air-sea fluxes in the study area (Thomson and Girton,
2017;Thomson et al., 2018). The deployment occurred on the transit to NTAS approximately 45 NM to the
southwest of the buoy with the intent of leaving the Wave Gliders in the water throughout the length of the cruise.
They were remotely piloted from shore via an online portal to cross gradients in SST and ocean currents. Data were
available in near real time which helped guide their course. The Wave Gliders were equipped with surface
meteorological sensors (bulk winds, air temperature, relative humidity, pressure, and longwave and shortwave
radiation), sky cameras, wave motion sensors, downward looking ADCPs for currents, and CTDs at 1 and 8 m depth
for conductivity (salinity) and temperature measurements at 1 and 8 m depth. Measurements were collected during
20-min bursts every 30 min. Final data products are 60-min averages of high-resolution raw data within each hour,
time-stamped at the beginning of the hour. Instrumentation onboard the Wave Gliders is listed in Table 7.

Wave Glider 245 was recovered, repaired, and redeployed on Jan 30. Telemetered data suggested that the humidity
sensor had malfunctioned. When recovered, it was found that the radiometers and their entire mounting pole were
gone, water was inside the data logger housing, the Airmar meteorological sensor and light were broken, and the
Vaisala meteorological sensor was destroyed. The radiation measurements lasted approximately one week into the
deployment. The Wave Glider was redeployed with spare Vaisala and Airmar meteorological sensors but no
radiometer. Wave Glider 245 was recovered for the final time on Feb. 7 because it was experiencing navigation
problems that could have endangered the vehicle. Wave Glider 247 sampled from Jan. 9 to Feb. 11. On Jan. 31,
Wave Glider 247 was inspected with the ship at close range after finding Wave Glider 245 damaged the day before.
The meteorological sensors were found to be in good condition but the radiometers had detached and were being
dragged by wires on the port side of vehicle. A small boat was deployed to clip the radiometer wires and take the
instruments back to the ship.
**Table 7**. Instrumentation onboard the Wave Gliders for the measurement of atmospheric and seawater parameters.
Data were collected during bursts lasting 20 min at the top of each hour. Measurements were collected during 20-
min bursts every 30 min. Final data products are 60-min averages of high-resolution raw data within each hour and
time-stamped at the beginning of the hour.

| Instrument | Measured quantity | Height (+) Depth (-) (m) | Raw sampling interval |
|---|---|---|---|
| Airmar 200WX | Wind velocity (true and relative), GPS position, COG, SOG, magnetic heading, temp, pressure, pitch and roll | +1.3 | 1 sec |
| Vaisala WXT530 | Wind velocity, air temperature, pressure, relative humidity, rain rate | +1 | 1 sec |
| Kipp Zonen CMP3 pyranometer | Short wave radiation (300-2800 nm) | +1 | 5 sec |
| Kipp Zonen CGR3 pyrgeometer | Long wave radiation (4200-4500 nm), temperature of sensor | +1 | 5 sec |
| GPSWaves/Microstrain 3DM-GX3-35 GPS/AHRS | Directional (2D) wave spectra, and standard bulk wave parameters of height, period, direction | 0 | 0.25 sec |
| Aanderaa 4319 | Conductivity, temperature | -0.24 | 2 sec |
| RDI Workhorse Monitor 300kHz ADCP | Ocean current profiles with 4 m vertical resolution | data between -6 to -100 m | 1 second pings, ensemble averages recorded every 2 min |
| Seabird GPCTD+DO | Conductivity, temperature, depth, dissolved O2 | -8 m | 10 sec |

**2.7. SWIFT measurements**
Drifting with ocean currents and winds, the SWIFTs (Surface Wave Instrument Floats with Tracking) offered a
Lagrangian view of the near-surface ocean and atmospheric properties, ocean waves and currents, bulk air-sea
fluxes, and cloud features (Thomson, 2012;Thomson et al., 2019). Instrumentation onboard SWIFTs v4 and v3 is
listed in Tables 8 and 9, respectively. Six SWIFT drifters were deployed in two SE-NW lines across gradients in
SST and ocean surface currents – once during Leg 1 and once during Leg 2. These gradients were identified with
satellite MUR v4 SST daily plots and the ship's underway thermistor, thermosalinograph, and ADCP. Two version 3
(serial number 16 and 17) and four version 4 (serial number 22, 23, 24, and 25) SWIFTs were deployed. All had
bulk meteorological sensors (winds, air temperature and pressure on all models, plus relative humidity on the v4
models), sky cameras, and CTD sensors at 0.3 m depth for measuring temperature and conductivity (salinity). The
v3 models also had conductivity and temperature sensors at 1.1 m depth. The v3 SWIFTs measured ocean
turbulence in the upper 0.62 m. The v4 SWIFTs measured ocean turbulence in the upper 2.64 m. Both versions had
ADCPs that measured vertical profiles of currents down to 20 m. The SWIFTs sampled high resolution bursts of
data for 8 min at the top of each hour. These data were archived on board the vehicle for final processing once
recovered. The 8-min data segments and platform location were also averaged and reported via Iridium satellite
telemetry each hour for monitoring purposes. SWIFT locations were also tracked in real time using the AIS ship
traffic system (local VHF radio signals). The SWIFTs were deployed for 8 days during Leg 1 (Jan. 14 to 22) and 13
days during Leg 2 (Jan. 30 to Feb. 11).

**Table 8**. Instrumentation onboard the version 4 SWIFTs (serial numbers 22, 23, 24, 25) for the measurement of
atmospheric and seawater parameters. Measurements were collected during 8-min bursts at the beginning of each
hour. Final data products are 8-min averages of high-resolution raw data, time-stamped at the beginning of each
hour.

| Instrument | Measured quantity | Height (+) Depth (-) (m) | Raw sampling interval |
|---|---|---|---|
| Vaisala WXT530 | Wind velocity, air T, barometric pressure, relative humidity, rain rate | 0.5 | 1 sec |
| Camera | 320 x 240 JPEG cloud images | 0.2 | 4 sec |
| SBG Ellipse GPS/AHRS | Directional (2D) wave spectra, and standard bulk wave parameters of height, period, direction | 0 | 0.2 sec |
| Nortek Signature 1000 ADCP with AHRS | | -0.3 to -2.64 | 0.25 sec |
| | Turbulent kinetic energy dissipation rate profiles with 0.04 m vertical resolution | | |
| | Ocean current profiles with 0.5 m vertical resolution | -0.35 to -20 | 0.25 sec |
| | 3-D motion and heading data | 0 | 0.25 sec |
| Aanderaa 4319 | Conductivity (salinity), temperature | -0.3 | 2 sec |


**Table 9**. Instrumentation onboard the version 3 SWIFTs (serial numbers 16 and 17) for the measurement of
atmospheric and seawater parameters. Measurements were collected during 8-min bursts at the beginning of each
hour. Final data products are 8-min averages of high-resolution raw data and time-stamped at the beginning of the
hour.

| Instrument | Measured quantity | Height (+) Depth (-) (m) | Raw sampling interval |
|---|---|---|---|
| Airmar 200WX | Wind velocity, GPS position, COG, SOG, magnetic heading, air temperature and pressure, pitch and roll | 0.8 | 1 sec |
| Camera | 320 x 240 JPEG cloud images | 0.2 | 4 sec |
| GPSWaves/Microstrain 3DM-GX3-35 GPS/AHRS | Directional (2D) wave spectra, standard bulk wave parameters of height, period, direction | 0 | 0.25 sec |
| Nortek Aquadopp ADCP | Turbulent kinetic energy dissipation rate profiles with 0.04 m vertical resolution | -0.02 to -0.62 | 0.25 sec |
| | Ocean current profiles with 0.5 m vertical resolution | -0.65 to -20 | 0.25 sec |
| Aanderaa 4319 | Conductivity (salinity), temperature | -0.50 | 2 sec |
| Aanderaa 4319 | Conductivity (salinity), temperature | -1 | 2 sec |


## 2.8. Saildrone measurements

NOAA sponsored two Saildrones for the ATOMIC campaign to obtain high quality multiscale air-sea fluxes (Zhang et al., 2019) in two different regimes. Both were launched from Bridgetown, Barbados on Jan. 12, 2020. Saildrone SD1063 focused on the large ocean eddies southeast of BCO, where the North Brazil Current Rings propagate northwestward toward Barbados. Saildrone 1064 sampled in Trade Wind Alley along the leg between BCO and NTAS. In addition, Saildrone 1064 coordinated sampling with the *RV Ronald H. Brown*, remote sensing from research aircrafts, NTAS, Wave Gliders, and SWIFTs. Saildrones 1063 and 1064 were equipped to measure near surface ocean temperature and salinity, upper-ocean current profiles (6m-100m), surface air temperature, humidity, pressure, wind direction and speed, wave height and period, short- and long-wave radiation, and cloud images (Table 10). This system enabled calculation of the bulk latent heat flux and direct turbulent fluxes of momentum and sensible heat. Six thermistors were strapped on the keel to measure the surface layer stratification. Onboard data processing included averaging and motion correction. One-minute averages (5-minute average for ADCP current) were telemetered in real time, while high resolution data were downloaded after the Saildrones returned to U.S. During the 1-month ATOMIC intensive observation period of Jan. 12 to Feb. 12, Saildrone 1064 continuously measured air-sea interaction processes between BCO and NTAS and sailed 1777 nautical miles. After ATOMIC, the Saildrones continued their observations until July 16 and then sailed back to the U.S. arriving in Newport, RI on August 30, 2020.

Three additional Saildrones were piloted by a NASA-funded effort. These data and their details are posted at https://podaac.jpl.nasa.gov/dataset/SAILDRONE_ATOMIC, https://doi.org/10.5067/SDRON-ATOM0.

## 2.9. RAAVEN UAS measurements

The University of Colorado operated a small remotely-piloted aircraft system (RAAVEN) from Morgan Lewis Beach on the northeastern shore of Barbados between Jan. 24 and Feb. 16. The miniFlux payload flew onboard the RAAVEN UAS (de Boer et al., 2020). Flights conducted during this campaign targeted the thermodynamic and kinematic structure of the lower atmosphere, with sampling occurring between the surface and 1 km altitude. The vast majority of the flights were focused on the sub-cloud layer, with extended sampling conducted at cloud base and a sequence of set altitudes within the sub-cloud layer. Included in these flights were regular sampling intervals at 20 m above the ocean surface to collect information on turbulent surface fluxes of heat and momentum. Most flights were conducted in the near-coastal zone between 0 and 2 km from the coastline. MiniFlux sensors included a multihole pressure probe (MHP); fine-wire array; IR thermometers; pressure, temperature and humidity sensors similar to those used in radiosondes and dropsondes; redundant pressure, temperature, and humidity probes; and an inertial navigation system.

**Table 10**. Instrumentation onboard the NOAA sponsored Saildrones. 1-minute averages (5-minute average for
ADCP current) were telemetered in real time except where noted below. Final data products are 1-min averages of
high resolution raw data.

| Instrument | Measured quantity | Height (+) Depth (-) (m) | Raw sampling interval |
|---|---|---|---|
| Gill WindMaster 1590-PK | Wind velocity (true and relative), GPS position, COG, SOG, magnetic heading, temp, pressure, pitch and roll | +5.2 | 0.1 sec |
| Rotronic Hygroclip 2 | Air temperature, relative humidity | +2.3 | 1 sec |
| SPN1 Delta-T Sunshine pyranometer | Short wave radiation | +2.8 | 0.2 sec |
| Eppley Precision Infrared Radiometer (PIR) | Long wave radiation, temperature | +0.8 | 1 sec |
| VectorNav VN300 DualGPS aided IMU (Wing) | GPS position, COG, SOG, magnetic heading, pitch and roll (motion correction for WindMaster and SPN1) | +2.575 | 0.05 sec |
| VectorNav VN300 DualGPS aided IMU (Hull) | Wave height and wave period and motion correction for ADCP currents | +0.34 | 0.05 sec |
| LICOR LI-192SA | Photosynthetically Active Radiation (PAR) | +2.6 | 1 sec |
| WET Labs ECO-Fluorometer | Chlorophyll-a (experimental) | -0.5 | 1 sec |
| RBR C.T.ODO.chl-a | Conductivity, temperature, dissolved O2, Chl-a (experimental) | -0.53 | Inductive CTD |
| Teledyne RDI Workhorse 300kHz ADCP | Ocean current profiles | -6 to -100 | 1 sec, 5 min avg. sent via telemetry |
| Heitronics CT15.10 | Skin seawater temperature (experimental) | 0 | 1 sec |
| Vaisala PTB 210 | Barometric pressure | +0.2 | 1 sec |
| 4 Cameras | Cloud image | Upward, sideways, downward | Every 5 min, telemetered every 30 min |
| Seabird SBE57 temperature loggers | Temperature | -0.3, -0.6, - 0.9, -1.2, -1.4, -1.7 | 1 sec, 1 min avg. not telemetered |
| Seabird SBE37 CTD+DO | Conductivity, temperature, depth, dissolved O2 | -0.5 | Pumped, burst sampled 10 sec for 1min/5 min |


**2.10. SVPS drifter measurements**


Though not deployed from the *R/V Ronald H. Brown*, the ATOMIC field campaign and its archive also includes a
dataset of 9 SVPS type surface ocean drifters deployed by NOAA AOML (Surface Velocity Program Salinity,
(Centurioni et al., 2015;Hormann et al., 2015)). These were deployed from the EUREC[4]A ship *RV L'Atalante* 50 to
150 NM from the South American coast, between 6°N and 10°N, the so-called Boulevard de Tourbillons (Eddy
Boulevard), where North Brazil Current Rings translate northwestward (Fig. 1b). The purpose of these drifters was
to measure air-sea interaction, ocean properties, and atmospheric variability amidst ocean eddies and low-salinity
plumes from a Lagrangian perspective. During ATOMIC the SVPS drifters measured air pressure and relative wind
at 0.5 m height. They also measured ocean salinity and temperature (0.3, 5, 10 m depth, with a duplicate T sensor at
0.3 m), and ocean velocity representative of water located between 11-19 m depth and centered at 15 m. The drifter
was equipped with a drogue centered at 15 m in the form of a long vertically-oriented holey sock. The drogue's full
extent spanned a depth of 11.34 m to 18.66 m. Therefore, currents calculated from the drifter location are
representative of currents between these depths. Bulk wind stress and the bulk drag coefficient were estimated from
these data using COARE 3.6. Data records began at different times and locations to sample different ocean features.
Four drifters started on Jan. 23, 1 drifter on Jan. 26, 4 drifters on Feb. 2, and 1 drifter on Feb 4. The drifters exited
the ATOMIC/EUREC4A region on about Apr. 29, which marks the end of this archived ATOMIC dataset. After
this date, data were still being reported from some sensors and can be accessed by contacting the PI (Table 11). The
drifter sensors sampled every 90 sec and then computed averages over 30 min. The averaged data were transmitted
to land via satellite telemetry. The position and time data were instantaneous every 30 min. Ten total drifters were
deployed but GPS did not work on one so that dataset is not posted.

**2.11. BACO aerosol measurements**

Size resolved cloud condensation nuclei (CCN) number concentrations were measured with a custom-made
differential mobility analyzer (DMA) for size selection connected to a Droplet Measurement Technologies (DMT)
CCNC-100 and a GRIMM 5.412 CPC. Aerosol number size distributions were made with a Scanning Mobility
Particle Sizer (SMPS) (GRIMM 5.420) with a diameter range of 0.10 to 1.094 μm. Measurements were made from
an isokinetic aerosol inlet located at roughly 47 m.a.s.l.

**2.12. BCO measurements**

BCO is at a height of 25 m.a.s.l. Meteorological sensors (Vaisala WXT-520) were mounted at 4 m.a.g.l.  BCO
launched 182 radiosondes. Data from the sondes were merged into the EUREC[4]A sounding network (Stephan et al.,
2020). A Lufft ceilometer CHN 15k NIMBUS was used for the determination of cloud base height.

**3. Overview of meteorological and surface seawater conditions sampled**

During ATOMIC, the *RV Ronald H. Brown* primarily operated in Trade Wind Alley, north of 12.5°N between
~56°W and NTAS (Fig. 1a). During the boreal winter, near-surface winds from the northeast carry air masses from
NTAS to BCO in about 1.5 d. Positioning the *RV Ronald H. Brown* in Trade Wind Alley allowed for sampling of
atmosphere and ocean conditions from the surface in between NTAS and BCO. Winds were fairly steady throughout
the cruise with an average speed (10 m) of $8.3 \pm 2.1$ m sec$^{-1}$ and direction of $70 \pm 21°$ (Fig. 4a). Air temperature (10
m) ranged between 22.7 and 27.9°C and averaged $25.7 \pm 0.61$ °C. RH averaged $71 \pm 4.7\%$ (Figure 4b). Radiosondes
launched within Trade Wind Alley revealed dryer conditions in the lower and middle troposphere compared to
observations made to the south in the Boulevard de Tourbillons which paralleled the coast of South America (Fig.
1b). Stephan et al. (2020) attribute the difference to more frequent periods of a deep moist layer and deeper
convection to the south.

Rain rate was measured by three instruments during the cruise located at different places on the ship. Two Parsivel
disdrometers were located on the port rail on the O3 deck, an ORG-815 DA optical range gauge was located on the
mast tower, and a Vaisala WXT536 was mounted on top of an aerosol sampling van on the O2 deck. Although
instruments and locations were not identical, a coherent picture of rain occurrence emerges with frequent events
**Figure 4.** Time series of bow mast measurements of a) wind speed and direction, b) air temperature and relative
humidity all adjusted to 10 m height using the COARE 3.6 bulk model. Also shown are c) rain rate measured with
three different instruments, d) dust and elemental carbon mass concentration for particles with aerodynamic
diameters less than 10 μm, and e) skin seawater temperature from the sea snake and downwelling longwave
radiation.

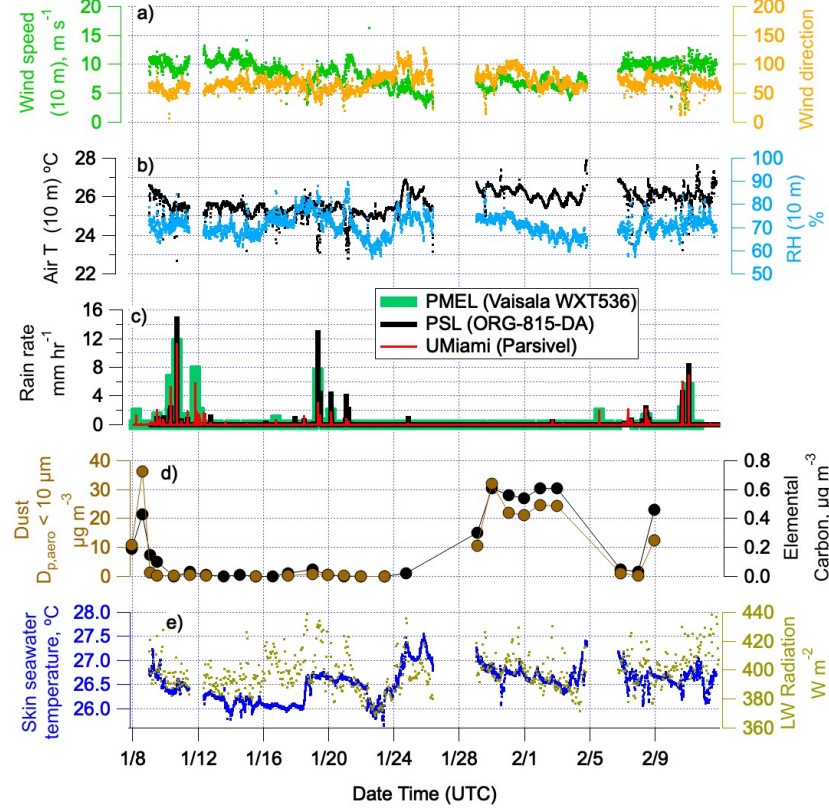



between Jan. 9 and 12; Jan. 19 and 21; and Feb. 8, 10, and 11 (Figure 4c). January rain events were associated with a
stationary front extending along 20°N from the east into Barbados. February events occurred as an Atlantic ridge
progressed eastward inducing strong winds and scattered showers.

One unique feature of the atmospheric conditions during ATOMIC was the occurrence of high concentrations of
dust in the boreal winter. Dust concentrations have long been documented to increase each summer in the Caribbean
due to transport from Africa (Prospero and Mayol-Bracero, 2013). A layer of warm, dry air above the marine
boundary, known as the Saharan Air Layer (SAL), extends from Africa to North America during the summer which
leads to relatively long aerosol residence times and efficient transport of dust between the two continents (Petit et
al., 2005;Carlson and Prospero, 1972). Factors contributing to dust transport to the Caribbean during the winter are
not as well understood but have been shown to correlate with the southward movement of the Intertropical
Convergence Zone (ITCZ) which affects near-surface northeasterly winds over North Africa (Doherty et al., 2012).
As a result, the SAL occurs at lower altitudes and more southern latitudes in the winter (Tsamalis et al., 2013;Liu et
al., 2012).

Filter measurements of particulate Al, Si, Ca, Fe, and Ti onboard the *RV Ronald H. Brown* were used to derive dust
concentrations (Malm et al., 1994). As shown in Fig. 4d, elevated dust concentrations were observed at the
beginning of Leg 1 (Jan 8 00:00 to Jan. 9 12:00) and two more times during Leg 2 (Jan. 29 12:00 to Feb. 3 19:00
and Feb. 9 00:00 to Feb. 11 12:00). Dust concentrations were still elevated when aerosol sampling was halted on
Feb. 3 and Feb. 11. Elemental carbon (EC) concentrations were enhanced during these same periods indicating
transport of biomass burning along with the dust. The NASA Fire Information for Resource Management System
(FIRMS) satellite product indicated a wide swath of fires over North Africa during January and February of 2020
(https://earthdata.nasa.gov/earth-observation-data/near-real-time/firms).

The ATOMIC study area was characterized by warmer skin seawater temperatures nearer to Barbados (west of
~55°W) due, in part, to the North Brazil Current (NBC) that transports South Atlantic warm water along the coast of
Brazil and into the northern hemisphere, separating from the coast around 6° to 8°N. Occasionally the NBC curves
back on itself and pinches off warm eddies that move further north and into the Caribbean Sea (Fratantoni and
Glickson, 2002). The coolest skin seawater temperatures were encountered in the vicinity of the NTAS and MOVE
operations on the most northeastern portion of the cruise track between Jan. 12 and 16 (Fig. 1a and Fig. 4e). A
second period of low skin seawater temperatures coincided with sustained relatively low longwave downwelling
radiation on Jan. 22 and 23 (Fig. 4e) although causes of the low temperatures have yet to be determined.

**4. Inter-platform data comparisons**

Times when the *RV Ronald H. Brown* was in close proximity to or upwind of other sampling platforms are listed in
Table 3. These periods provide the potential for inter-platform comparisons for data quality checks or scientific
purposes. Inter-platform comparisons reported here include 1) NTAS moorings and the ship (seawater and
atmospheric parameters), 2) Saildrone 1064 and the ship (seawater and atmospheric parameters), 3) BCO and the
ship (atmospheric parameters), 4) BACO and the ship (aerosol properties), and 5) BCO, the ship, and RAAVEN
UAS  (cloud base height). These comparisons were done to evaluate consistencies in the measurements. Resolving
identified inconsistencies will be the subject of future research.

**4.1. Comparison of seawater parameters**

4.1.1. Onboard RHB. No significant offsets or biases were found among the independently calibrated subsurface
temperature measurements onboard *RV Ronald H. Brown*. Measurements from the ship's CTD, uCTD, PSL sea
snake, and ship's underway thermosalinograph and thermistor were similar. After correcting for a small bias found
in the uCTD salinity, no significant difference was found among the different salinity measurements recorded.

4.1.2. NTAS – RHB. Four CTD casts with the ship's rosette were conducted to compare to the NTAS moorings'
upper ocean measurements between Jan. 12 and 15. The ship was 3 NM southwest of the NTAS-18 mooring anchor
on Jan. 12 and 13 and 3.8 NM northwest of the NTAS-17 mooring anchor on Jan. 15 (Table 3 and Fig. 5a). With an
anchor radius watch circle of ~ 2 NM for each buoy, the ship and buoys were within 0.25 to 3 NM of each other.
NTAS measurements of temperature and salinity at 5 depths (10, 25, 40, 55, and 70 m) are overlaid onto data from
the ship's CTD in Figure 5. Absolute differences (NTAS – RHB) in temperature are less than 0.1°C for all depths of
the three casts conducted on Jan. 12 and 13 except for the last cast during that period (Fig. 5j). For the most part, the
salinity comparisons show good agreement for the Jan. 12 and 13 casts with absolute differences at depths between
10 and 40 m being less than 0.03 (Fig.5 k). Exceptions occurred at lower depths due to strong vertical gradients.

The comparison from Jan. 15 shows significant differences for both temperature (Fig. 5j) and salinity (Fig. 5k)
likely due to horizontal gradients. Satellite derived sea surface salinity and SST for this day indicate that NTAS and
the ship were located in a frontal region with the ship in warmer and saltier surface water to the north of NTAS. The
sign of the absolute differences (NTAS – RHB) in temperature and salinity varied with depth. The ship's ADCP
revealed vertical structure in the currents consistent with the sign of observed absolute differences at the surface
versus lower depths.

**4.2. Comparison of atmospheric parameters**

4.2.1. NTAS – RHB

Atmospheric parameters (temperature, relative humidity, specific humidity, wind speed, pressure, rain rate, and
longwave downwelling radiation) measured onboard the NTAS buoys and the *RV Ronald H. Brown* were compared
when the platforms were within 3 NM of each other between Jan. 10 and 15 (Table 3). Measurements from NTAS-
17 and NTAS-18 were combined into one data set for the comparison. Based on 1-hr averaged data, 59 samples
were available for comparison.

Wind speed, temperature, and specific humidity from both platforms were adjusted to a height of 10 m. Absolute
differences (NTAS – RHB) were positive for temperature (Fig. 6a), RH (Fig. 6b), specific humidity (Fig. 6c), and
wind direction (Fig. 6d). These differences, however, were within either reported accuracies of the instrumentation
or within the range reported for a previous 24-hr *RV Ronald H. Brown* – Stratus 4 buoy comparison (Colbo and
Weller, 2009). Absolute differences (NTAS – RHB) were negative for wind speed, pressure, rain rate, and longwave
downwelling radiation although all differences were within accuracies of the instrumentation or within the range
reported by Colbo and Weller (2009).




**Figure 5.** Comparison of upper ocean measured parameters from the NTAS 18 mooring and the *RV Ronald H.*
*Brown* on Jan. 12, 13, and 15 with a) location of NTAS-18 mooring anchor. The NTAS buoys were about 2 NM
downwind (SW) of the anchor so the CTD and mooring measurements were within 0.5 to 3 NM of each other. Also
shown are b – e) temperature, f – i) salinity, and j – k) absolute differences and root mean square differences (rmsd)
for temperature and salinity, respectively. Number of samples = 4.

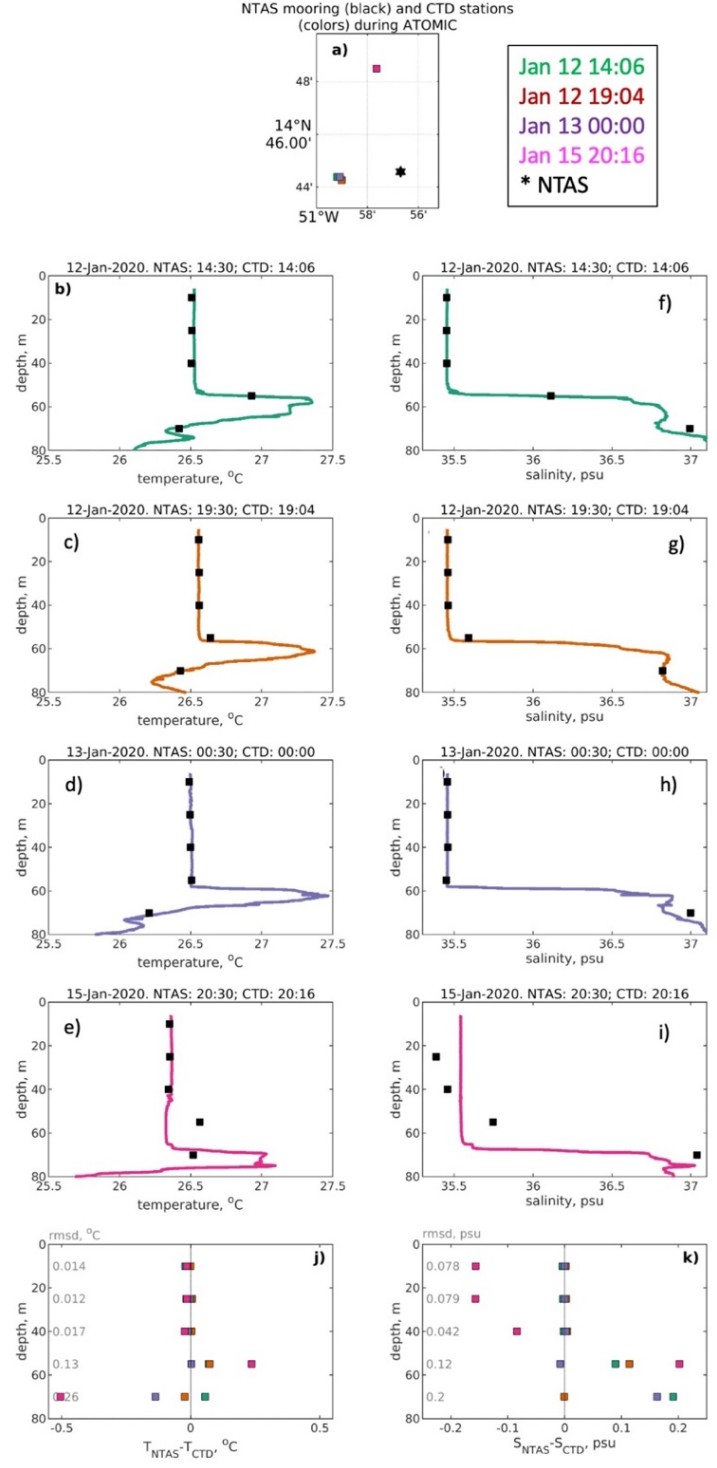


**Figure 6.** Comparison of meteorological parameters measured onboard the NTAS buoy and the *RV Ronald H. Brown* (RHB) when the platforms were between 0.25 and 3 NM apart between Jan. 10 and Jan. 15 including a) atmospheric temperature (10 m), b) relative humidity (10 m), c) specific humidity (10 m), d) wind direction, e) wind speed (10 m), f) atmospheric pressure (10 m), g) rain rate, and h) longwave downwelling radiation. The averaged of absolute differences (NTAS – RHB) and root mean square differences (rmsd) are reported in the inset table. Number of samples based on 1 hr averaged data = 59.

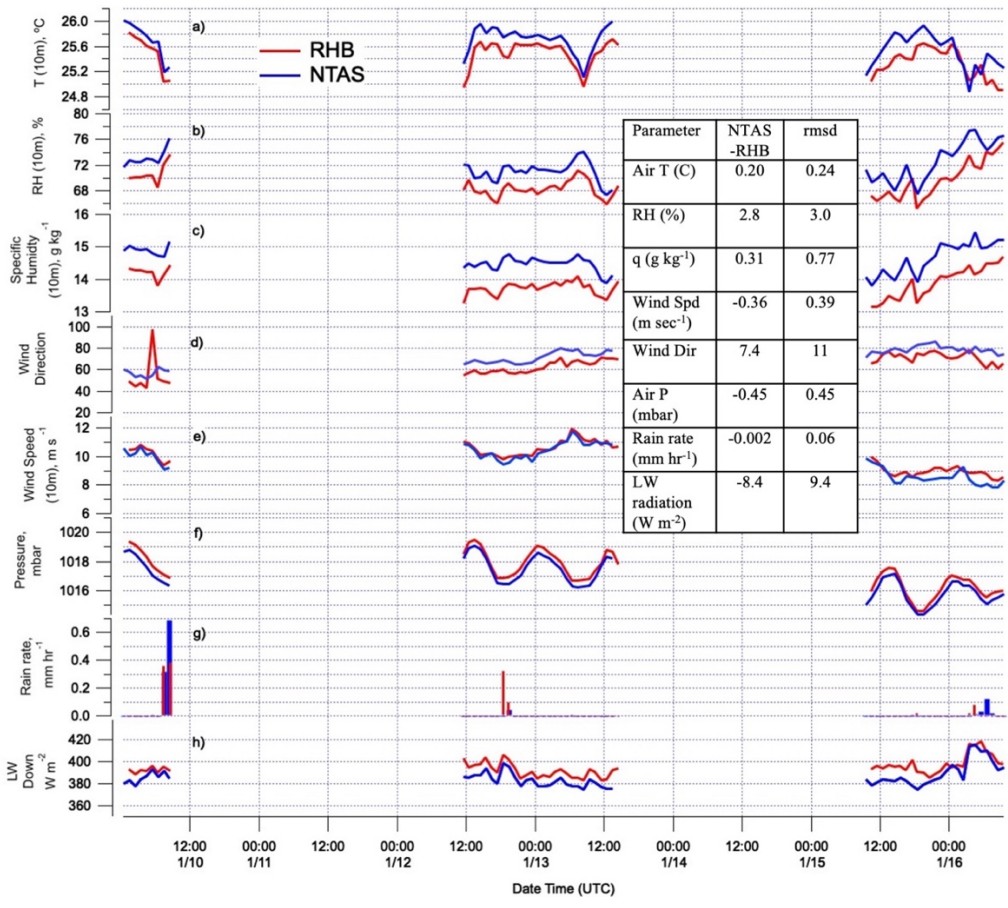

4.2.2. BCO – RHB

The Barbados Cloud Observatory (BCO) is located at Deebles Point on the eastern coast of Barbados. Atmospheric parameters (temperature, RH, wind direction and speed, pressure, and rain rate) were compared between BCO and the ship during the period the ship was 20 NM east and upwind of the observatory (Jan. 24 18:20 to Jan. 25 23:40) (Table 3). BCO meteorological sensors were located at 30 m.a.s.l. and were not adjusted to a height of 10 m due to uncertainties in adjusting overland measurements (BC0) with surface fluxes from the *RV Ronald H. Brown.* Based on 10-min averaged data, 177 samples were available for comparison.

The average of the absolute difference (BCO – RHB) in temperature over the entire period was larger than instrumental accuracies (Fig. 7a). The largest difference was observed after 12:00 UTC (08:00 local) indicating

relatively more warming of the sensor and/or atmosphere at BCO due to diurnal heating of the land surface. Even
with differences in temperature, RH values from the two platforms agreed well with the exception of the end of the
period. The ship observed an abrupt change in temperature and RH on Jan. 25 at 19:30 (Fig. 7a,b) suggesting that
the platforms were in different air masses. Wind direction agreed well between platforms (Fig. 7c) but the average
of the absolute differences (BCO – RHB) in wind speed (Fig. 7d) and pressure (Fig. 7e) were larger than
instrumental uncertainties. One rain event occurred during the comparison. It was observed on Jan. 24 on the ship
and 30 minutes later at BCO with observed rain rates of 1.2 and 3.5 mm hr$^{-1}$, respectively (Fig. 7f).

**Figure 7.** Meteorological parameters measured during the *RV Ronald H. Brown* (RHB) and the Barbados Cloud
Observatory (BCO) comparison (Jan. 24 18:20 to Jan. 25 23:40) when RHB was 20 NM due east of BCO.
Parameters include a) atmospheric temperature, b) relative humidity (RH), wind direction, wind speed, atmospheric
pressure, and rain rate. The average of the absolute differences (BCO – RHB) and root mean square differences
(rmsd) are reported in the inset table. BCO meteorological sensors were located at 30 m.a.s.l. and were not adjusted
to a height of 10 m. Number of samples based on 10 minute averaged data = 177.

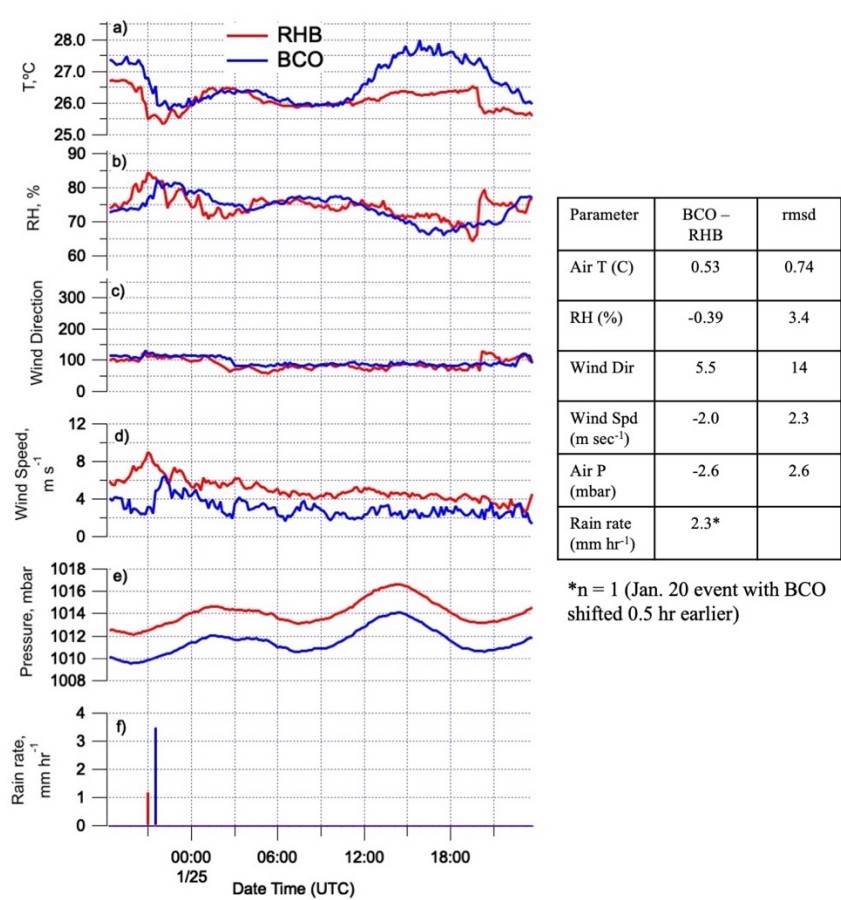

| Parameter | BCO – RHB | rmsd |
|---|---|---|
| Air T (C) | 0.53 | 0.74 |
| RH (%) | -0.39 | 3.4 |
| Wind Dir | 5.5 | 14 |
| Wind Spd (m sec$^{-1}$) | -2.0 | 2.3 |
| Air P (mbar) | -2.6 | 2.6 |
| Rain rate (mm hr$^{-1}$) | 2.3* | |

*n = 1 (Jan. 20 event with BCO shifted 0.5 hr earlier)




4.2.3. SD1064 – RHB

Saildrone 1064 and the *RV Ronald H. Brown* were within 0.7 to 3.6 NM of each other between  Feb. 8 and 10.
Based on 10 min averaged data, 663 samples were available for the comparison. Air temperature, RH, and wind
speed adjusted to 10 m were used for the comparison. Skin seawater temperature was measured at a depth of 0.05 m
on the Saildrone and from the ship's Sea Snake. On average, skin seawater temperature agreed within 0.01°C,
atmospheric temperature within 0.12 °C, and RH within 1.9%. – all within the uncertainty of the measurements or
within the agreement observed between the NTAS buoy and the ship (see Sect. 4.2.1.) (Fig. 8a, b, c). At the end of
the comparison, ship measured seawater temperature at 0.05 m decreased, atmospheric temperature decreased, and
RH increased while Saildrone observed parameters remained steady even though the platforms were within 0.8 NM
of each other. These differences indicate the fine scale nature of structural differences in surface oceanic and lower
atmospheric conditions.

On average, agreement for wind direction and wind speed was not within instrumental uncertainties or the
agreement observed between the NTAS buoy and the ship due to spikes in the ship's measurements not observed by
the Saildrone (Fig. 8d, e). Atmospheric pressure agreed well with an absolute difference (SD0164 – RHB) of -0.27
mbar (Fig. 8f). The absolute difference in downward long wave radiation (SD0164 – RHB) was 4.4 W m$^{-2}$,
indicating a systemic offset (Fig. 8g).


**Figure 8.** Comparison of parameters measured onboard Saildrone 1064 (SD1064) and *RV Ronald. H. Brown* (RHB)
when the platforms were within 0.7 to 3.6 NM of each other between Feb. 8 and 10. Parameters include a) SST (SD
at -0.05 m and RHB Sea Snake), b) air temperature (10 m), c) RH (10 m), d) wind direction, e) wind speed (10 m),
f) atmospheric pressure (10 m), and g) longwave downwelling radiation. Absolute differences (SD1064 – RHB) and
root mean square differences (rmsd) are reported in the inset table. Number of samples based on 10 min averaged
data = 663.

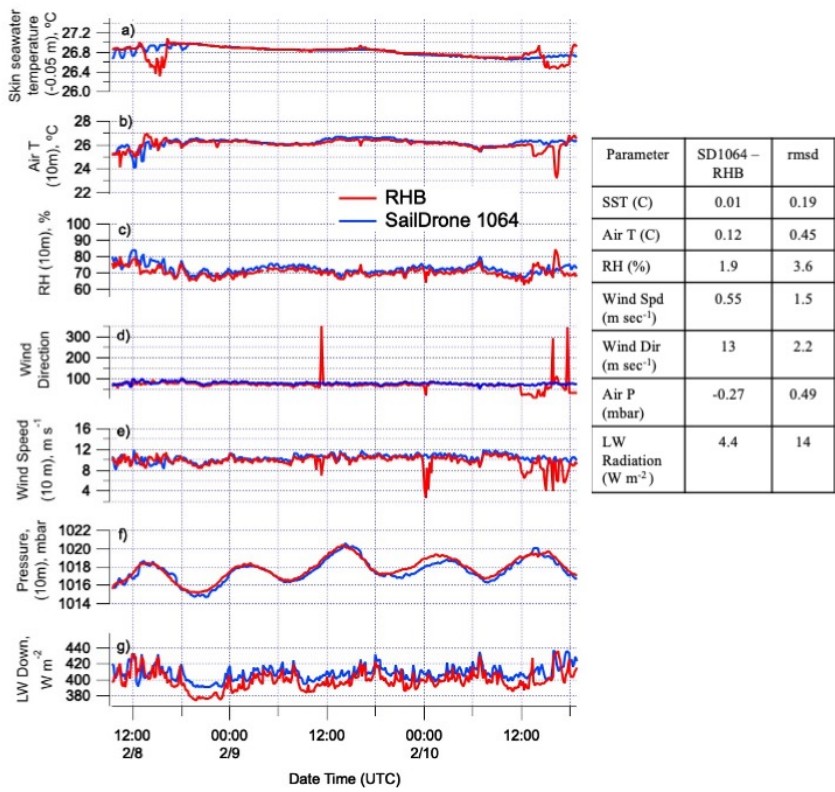

| Parameter | SD1064 – RHB | rmsd |
|---|---|---|
| SST (C) | 0.01 | 0.19 |
| Air T (C) | 0.12 | 0.45 |
| RH (%) | 1.9 | 3.6 |
| Wind Spd (m sec⁻¹) | 0.55 | 1.5 |
| Wind Dir (m sec⁻¹) | 13 | 2.2 |
| Air P (mbar) | -0.27 | 0.49 |
| LW Radiation (W m⁻²) | 4.4 | 14 |


**4.3. Comparison of aerosol and cloud parameters**

4.3.1. BACO – RHB – aerosol parameters

The Barbados Atmospheric Chemistry Observatory (BACO) is located at Ragged Point, 400 m across a cove from BCO. Total particle number concentration (CN), cloud condensation nuclei (CCN) concentration at 0.4% supersaturation, and particle number size distributions were compared between BACO and the ship when the ship was 20 NM east and upwind of BACO (Jan. 24 18:20 to Jan. 25 23:40) (Table 3). Details on the *RV Ronald H. Brown* aerosol measurements are shown in Table 5 and details on CCN calibration and measurements are provided in Quinn et al. (2019). Details on BACO CCN calibrations and measurements are provided in Pöhlker et al. (2018).

CN and CCN concentrations are shown in Fig. 9 from the time when BACO measurements began (Jan. 22 00:16) to when the ship's measurements ended (Feb. 9 20:20). Based on CN concentrations below 300 $cm^{-3}$, both platforms encountered clean marine conditions until ~Jan. 29 at 12:00. Subsequent enhanced concentrations of both CN and CCN correspond to periods when dust and biomass burning reached the study area after transport from Africa (Fig. 4d) as observed in related earlier studies (Wex et al., 2016). The coherence of CN and CCN between the platforms, even when separated by 4 degrees of longitude, indicates a broadscale dust event.

**Figure 9.** Aerosol parameters measured onboard the *RV Ronald H. Brown* (RHB) and at Barbados Atmospheric Chemistry Observatory (BACO) for the period of overlapping measurements. The rectangle indicates the comparison period (Jan. 24 18:20 to Jan. 25 23:40) when RHB was 20 NM due east of BACO. Parameters include a) total particle number condensation (CN) and b) cloud condensation nuclei concentration (CCN) measured at 0.4% supersaturation. The average of the absolute differences (BACO – RHB) and root mean square differences (rmsd) for the comparison period are reported in the inset table. Number of samples = 5.

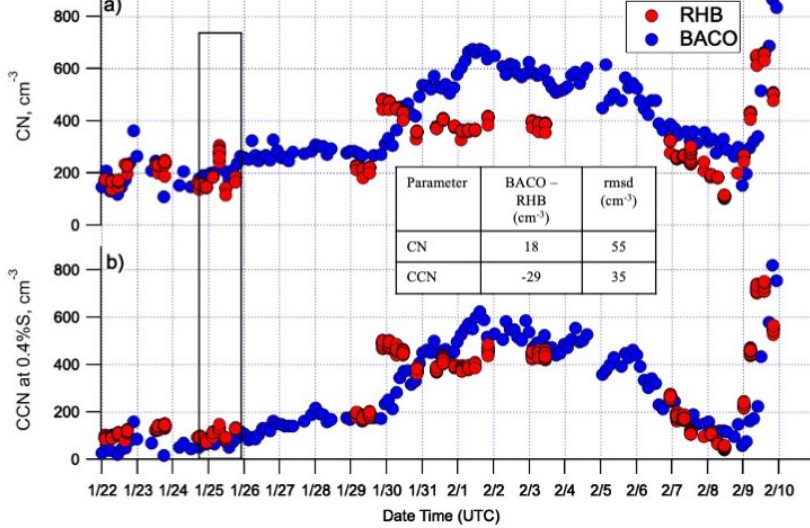

The comparison when the ship was 20 NM upwind of BACO is indicated by the rectangle in Fig. 9. CCN
concentrations were compared at a single supersaturation (S = 0.4%) which limited the number of samples to 5. The
absolute difference (BACO – RHB) was 18 cm$^{-3}$ for CN, which is less than 10% of the average CN concentration
during the comparison period and less than measurement uncertainties (Rose et al., 2008) (Fig. 9a). The difference
for CCN was -29 cm$^{-3}$ indicating the ship observed more CCN at S = 0.4% than BACO (Fig. 9b). However, this
difference is within the combined uncertainty of 30% for mono- and polydisperse CCN measurements.

Shipboard and BACO size distributions averaged over the length of the comparison were bimodal with Aitken
modal diameters of ~40 nm for both the ship and BACO and 130 and 170 nm for the accumulation mode for the
ship and BACO, respectively (Fig. 10). Differences in magnitude could be due to instrumental issues or local
aerosol sources at BACO.

**Figure 10.** Comparison of aerosol number size distribution measured onboard the *RV Ronald H. Brown* (RHB) and
at the Barbados Atmospheric Chemistry Observatory (BACO) during the comparison period (Jan. 24 18:20 – Jan. 25
23:40) when RHB was 20 NM to the east of BACO.

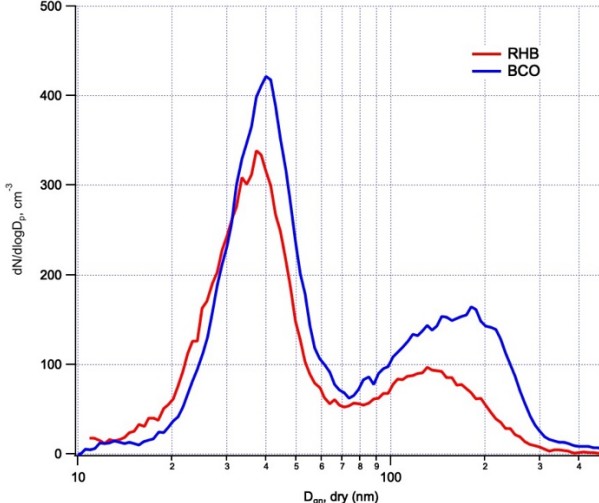

4.3.2. BCO – RHB – cloud base height

Cloud base height (CBH) was derived from three different measurements onboard the *RV Ronald H. Brown* – LCL
calculated from the stitched together RHB – BCO radiosonde record (equation 1), the ceilometer, and the microDop
lidar (Fig. 11a). Fifth and 10$^{th}$ percentile values of the lowest cloud scattered return were averaged over 10 min
intervals of the ceilometer and lidar data, respectively. The choice of percentile levels was tested to reduce inclusion
of scattering at the surface made by rain and scattering aloft from horizontally-sheared cloud edges. Higher altitude
ceilometer and lidar values that remain in this time series are not representative of cloud base due to the presence
and scattering by sheared edges or detrained portions of clouds that are separated horizontally from the locations of
cloud base. Dilution of surface parcels with drier air could also contribute to rising heights of the cloud base. Lowest
values from both the ceilometer and lidar track well with the LCL values derived from the radiosondes.
**Figure 11.** Comparison of Cloud Base Height (CBH) for a) legs 1 and 2 onboard *RV Ronald H. Brown* (RHB) based
on LCL calculated from the stitched together RHB – BCO radiosonde record (equation 1), the ceilometer, and the
Doppler lidar and b) for the RHB – BCO comparison period (Jan. 24 18:20 – Jan. 25 23:40) based on the BCO
ceilometer, LCL from BCO radiosondes, the RHB ceilometer and microDop lidar, and the RAAVEN UAS flown
from Morgan Lewis (30 km north of BCO). Locations of the RAAVEN launch site, BCO, and RHB are shown in c).
The average of the absolute differences and root mean square differences (rmds) are shown in the table inset relative
to the RHB lidar-derived CBH. N indicates number of samples used in the comparison. *rmsd for RHB ceilometer –
RHB microDop lidar with CBH greater than 1000 m removed from comparison.

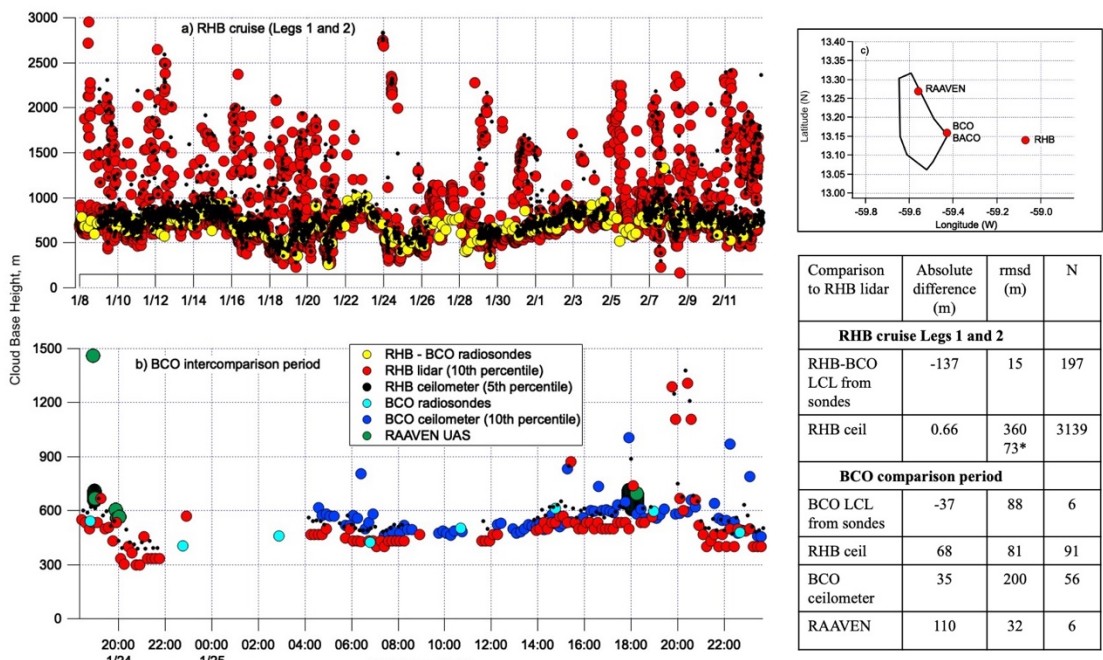


On average, the absolute difference between the LCL and lidar values (RHB-BCO LCL – RHB lidar) is -137 m due
to lidar scattering off of slightly higher altitude clouds. On average, the absolute difference between the RHB
ceilometer and microDop lidar (RHB ceilometer – RHB lidar) is 0.66 m indicating good agreement while the rmsd
value of 360 m reveals larger point to point differences. If CBHs from the ship's ceilometer and microDop lidar are
limited to values less than 1000 m, rmsd decreases to 73 m.

For the BCO comparison period (Jan. 24 18:20 – Jan. 25 23:40), CBHs were compared from the ship's ceilometer
and microDop lidar, BCO's ceilometer and LCLs from radiosondes, and the RAAVEN UAS miniFlux payload (Fig.
11b). The RAAVEN UAS flew from the eastern side of Barbados, 30 km north of BCO. Locations of the RAAVEN
launch site, BCO, and ship during the comparison are shown in Fig. 11c. Absolute differences in average values
between the BCO ceilometer and RHB microDop lidar (BCO ceilometer – RHB microDop lidar) and BCO LCLs
and RHB microDop lidar (BCO LCL – RHB microDop lidar) are around 35 m or 7% of the average sonde-derived
CBH. The absolute difference in average values between the RHB ceilometer and RHB microDop lidar (RHB
ceilometer – RHB microDop lidar) is slightly higher at 68 m.  RAAVEN values are slightly higher with an absolute
difference (RAAVEN miniFlux – RHB microDop lidar) of 110 m. Differences could be related to finer-scale
horizontal and vertical variability in boundary layer structure not readily-resolved by the measurements.

**5. Data availability**

All ATOMIC data sets discussed are publicly available at the NOAA PSL ATOMIC ftp server
(ftp://ftp2.psl.noaa.gov/Projects/ATOMIC/data/) (Quinn et al., 2020). Point of contact information and links to the
data sets are provided in Table 11. In addition, data have been submitted to NOAA's National Center for
Environmental Information (https://www.ncei.noaa.gov/) for Digital Object Identifiers (DOIs). The data will be
permanently and publicly available on the PSL ftp server and NCEI.

A readme file (README_ATOMIC_DATA.pdf) is available at ftp://ftp2.psl.noaa.gov/Projects/ATOMIC/data/
which describes the file structure of the ATOMIC folder and the content of the single files.

All of the datasets included in the discussion have been quality-controlled based on procedures implemented by the
individual research teams. Versioning also is based on protocols put in place by individual research teams. Details
can be found in the references listed in Table 11. Data are CF compliant. File name structure is:

<campaign_id>_<project_id>_<platform_id>_<instrument_id>_<variable_id>_<time_id>_<version_id>.nc.

An example for data collected from the ceilometer on the *RV Ronald H. Brown* is as follows. The name of the data
and link are:

EUREC4A_ATOMIC_RonBrown_Ceilometer_10min_20200109_20200212_v1.0.nc.

Metadata are embedded in the individual .nc files for each data set.


**Table 11**. Summary of data sets, links to data sets, point of contact information, and references for data collected
onboard the *RV Ronald H. Brown*, NTAS, Wave Gliders, SWIFTs, NOAA and NASA operated Saildrones, and
RAAVEN UAS during ATOMIC. Links in the table are for the ftp server
(ftp://ftp2.psl.noaa.gov/Projects/ATOMIC/data/). (Quinn et al., 2020). Data have been submitted to NOAA's
National Center for Environmental Information (https://www.ncei.noaa.gov/) for Digital Object Identifiers (DOIs)
and for permanent archiving. The data will be permanently and publicly available on the PSL ftp server, and NCEI.

| Platform | Data set | Data Link (preliminary FTP site location while NCEI DOIs are still being minted) | Point of Contact | Reference |
|---|---|---|---|---|
| All | ATOMIC | ftp://ftp2.psl.noaa.gov/Projects/ATOMIC/data/ | elizabeth.thompson@noaa.gov | Zuidema (2020) |
| RHB | Air-sea fluxes, ship navigation/location information, meteorological parameters, solar and infrared radiation, rain rate, subskin seawater T, skin seawater T (NOAA PSL) | ftp://ftp2.psl.noaa.gov/Projects/ATOMIC/data/rhb/met_sea_flux_nav/ | elizabeth.thompson@noaa.gov | Fairall et al. (1997);Fairall et al. (2003);Edson et al. (2013) |
| | ROSR skin seawater T (NOAA PSL) | ftp://ftp2.psl.noaa.gov/Projects/ATOMIC/data/rhb/ROSR/ | elizabeth.thompson@noaa.gov | |
| | Ceilometer (NOAA PSL) | ftp://ftp2.psl.noaa.gov/Projects/ATOMIC/data/rhb/ceilometer/ | elizabeth.thompson@noaa.gov | |
| | Disdrometer (rain rate, drop number, equivalent radar reflectivity) (U Miami) | ftp://ftp2.psl.noaa.gov/Projects/ATOMIC/data/rhb/disdrometer/ | pzuidema@rsmas.miami.edu | (Löffler-Mang and Joss, 2000) |
| | W-band radar (U Miami in partnership with NOAA PSL) | ftp://ftp2.psl.noaa.gov/Projects/ATOMIC/data/rhb/W-band-radar/ | pzuidema@rsmas.miami.edu elizabeth.thompson@noaa.gov | |
| | Sky camera (U Miami) | https://www.dropbox.com/sh/zejurecda70bilq/AABlLWgrEv1MDZ07yIE5TgWWa?dl=0 | pzuidema@rsmas.miami.edu | |
| | M-AERI skin seawater T, air humidity and temperature (U Miami) | ftp://ftp2.psl.noaa.gov/Projects/ATOMIC/data/rhb/M-AERI/ | pzuidema@rsmas.miami.edu gszczodrak@rsmas.miami.edu | Szczodrak et al. (2007) |
| | Doppler lidar (NOAA CSL) | ftp://ftp2.psl.noaa.gov/Projects/ATOMIC/data/rhb/doppler_lidar/ | alan.brewer@noaa.gov | Schroeder et al. (2020) |
| | Picarro water vapor isotopes (OSU/NCAR) | ftp://ftp2.psl.noaa.gov/Projects/ATOMIC/data/rhb/Picarro/ | david.noone@auckland.ac.nz | |
| | Picarro seawater isotopes | ftp://ftp2.psl.noaa.gov/Projects/ATOMIC/data/rhb/seawater_isotopes/ | david.noone@auckland.ac.nz | |
| | Meteorological and aerosol properties (NOAA PMEL) | ftp://ftp2.psl.noaa.gov/Projects/ATOMIC/data/rhb/atmos-chem/ | derek.coffman@noaa.gov | Bates et al. (2002) |
| | Radiosondes (OSU) | https://doi.org/10.5194/essd-2020-174 | simon.deszoeke@oregonstate.edu | Stephan et al. (2020) |
| | Underway CTD, uCTD (APL-UW) | ftp://ftp2.psl.noaa.gov/Projects/ATOMIC/data/rhb/UCTD/ | kdrushka@apl.uw.edu | Mojica and Gaube (2020) |
| | Ship rosette CTD (APL-UW) | ftp://ftp2.psl.noaa.gov/Projects/ATOMIC/data/rhb/CTD/ | kdrushka@apl.uw.edu | |
| | Ship ADCP (APL-UW) | ftp://ftp2.psl.noaa.gov/Projects/ATOMIC/data/rhb/ADCP/ | kdrushka@apl.uw.edu | |
| NTAS mooring | Meteorological parameters, air-sea fluxes, solar and infrared radiation; ocean currents, waves, conductivity, salinity, and temperature (WHOI) | ftp://ftp2.psl.noaa.gov/Projects/ATOMIC/data/NTAS/ | aplueddemann@whoi.edu | Weller (2018) |
| Wave Gliders | Air-sea fluxes, meteorological parameters, radiation; ocean currents, turbulence, waves, conductivity, and temperature (APL-UW) | ftp://ftp2.psl.noaa.gov/Projects/ATOMIC/data/wavegliders/ | jthomson@apl.washington.edu | Thomson and Girton (2017) |
| SWIFT drifter | Air-sea fluxes, meteorological parameters, radiation; ocean currents, turbulence, waves, | ftp://ftp2.psl.noaa.gov/Projects/ATOMIC/data/swift_drifters/ | jthomson@apl.washington.edu | Thomson et al. (2019) |

| | conductivity, and temperature (APL-UW) | | | |
|---|---|---|---|---|
| Saildrones (NOAA) | Air-sea fluxes, meteorological parameters, radiation; ocean currents, waves, conductivity, and temperature (NOAA PMEL) | ftp://ftp2.psl.noaa.gov/Projects/ATOMIC/data/saildrones_noaa/ | dongxiao.zhang@noaa.gov | Zhang et al. (2019) |
| Saildrones (NASA) | Air-sea fluxes, meteorological parameters, radiation; ocean currents, waves, conductivity, and temperature (NASA) | https://doi.org/10.5067/SDRON-ATOM0 | cgentemann@faralloninstitute.org | |
| SVPS drifters | Meteorological and ocean parameters, wind stress (NOAA AOML) | ftp://ftp2.psl.noaa.gov/Projects/ATOMIC/data/svp-s_drifters/ | greg.foltz@noaa.gov | Centurioni et al. (2015);Hormann et al. (2015) |
| RAAVEN miniFlux | Met parameters (Univ. Colorado) | ftp://ftp2.psl.noaa.gov/Projects/ATOMIC/data/CU-RAAVEN/ | gijs.deboer@noaa.gov | de Boer et al. (2020) |



As an example, the metadata for Cloud Base Height is:

long_name: cloud base height
standard_name: cloud_base_altitude
units: km
coverage_content_type: thematicClassification
instrument: ceilometer_instrument
platform: RonBrown
coordinates: time
cell_methods: time: point
valid_range: 0.0, 7.0
actual_range: 0.28, 6.86975
_FillValue: -9999.0
comment: Computed as the 5th percentile of cloud1, the height of first cloud layer detected, from 15 sec raw data
over this time period.


## 6. Summary

During ATOMIC, *in situ* and remote sensing measurements of oceanic and atmospheric properties and air-sea fluxes were made from the *RV Ronald H. Brown.* In addition, the NTAS mooring, radiosondes, SWIFTs, and Wave Gliders were deployed. Descriptions of the instrumentation onboard the ship and the deployed assets are provided along with the sampling strategy and day-to-day events. Atmospheric and oceanic conditions encountered during the cruise are described. Also detailed is how to access to all data collected. Comparisons were conducted with the NTAS moorings, Saildrone 1064, BCO, BACO, and the RAAVEN UAS. Data from inter-platform comparisons are presented to assess consistency in data sets. Resolving identified inconsistencies will be the subject of future research. The intention of the paper is to advance widespread use of the data by the ATOMIC and broader research communities.

**Author contributions.** P.K.Q. prepared the paper with the help of all co-authors. E.T. prepared data sets for archival on the PSL ftp server and at NCEI. D.J.C. prepared data for inclusion in the paper's figures. All authors participated in collecting and analyzing ATOMIC data.

**Competing Interests.** The authors declare that they have no competing interests.

**Acknowledgements.** We thank the crew of the *RV Ronald H. Brown* for their enthusiastic help and cooperation throughout the ATOMIC cruise and Dr. Edmund Blades and Peter Sealey for technical support at the BACO site. NOAA's Climate Variability and Predictability Program provided funding under NOAA CVP NA19OAR4310379, GC19-301, and GC19-305. The Joint Institute for the Study of the Atmosphere and Ocean (JISAO) supported this study under NOAA Cooperative Agreement NA15OAR4320063. Additional support was provided by the NOAA's Uncrewed Aircraft Systems (UAS) Program Office, NOAA's Physical Sciences Laboratory, and NOAA AOML's Physical Oceanography Division.  The NTAS project is funded by the NOAA's Global Ocean Monitoring and Observing Program (CPO FundRef number 100007298), through the Cooperative Institute for the North Atlantic Region (CINAR) under Cooperative Agreement NA14OAR4320158. We would like to thank Dr. David Farrell of the Caribbean Institute for Meteorology and Hydrology (CIMH) for his assistance with the organization of this campaign and Dr. Sandy Lucas of NOAA's Climate Program Office for her efforts that made ATOMIC and related outreach programs a success. This is PMEL contribution number 5172.

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
