# Peer review of "Measurements from the RV Ronald H. Brown and related 1 platforms as part of the Atlantic Tradewind Ocean-Atmosphere 2 Mesoscale Interaction Campaign (ATOMIC) 3 4 5 Patricia K. Quinn1, Elizabeth J. Thompson2, Derek J. Coffman1, Sunil Baidar3,4, Ludovic"

_Earth System Science Data, 2020_

## Referee Comment (RC1) · Anonymous Referee #1 · 28 Dec 2020

This paper describes recent measurements collected by several instrument platforms (ship, buoys, remotely operated ocean and aerial systems) in the vicinity of Barbados during 2020 to study shallow trade-wind clouds. Other papers describe other instrument platforms associated with ATOMIC and the corresponding EUREC4A campaign. The purpose of the paper is to present the types of measurements available as well as the status of particular instruments during their deployment. The intercomparisons of measurements (where appropriate) are particularly useful to provide users confidence in the dataset. As such, this paper will be useful as a reference to those wishing to

use the dataset for scientific analysis. I have checked the ftp and web sites to confirm that the data described is available to the public. While there does not appear to be any unique measurements, the combination of upper-ocean, near-surface, and lower atmospheric measurements from the platforms (and when combined with measurements described elsewhere) is a good dataset to advance the understanding of shallow tropical convective clouds as well as air-sea interactions and the marine boundary layer.

General Comments:

1) The paper is well-written and organized and I just have some specific comments to clarify the details. At times, the paper uses nautical jargon that may be difficult for some atmospheric sciences to understand. The authors should try to make the text generic enough for both atmospheric and ocean science communities.

2) There are instances in which instrumentation did not work – which is common for field campaigns. But it would be useful for the authors to elaborate a bit on how the missing planned data might have on the overall science objectives of the campaign.

3) The paper does a good job at describing the how sampling was conducted during the campaign, but in many places it does not provide any rational as to why the sampling strategy was conducted to serve the objectives of the campaign. A few sentences here and there will help the reader understand the reason for those choices and provide potential users the scientific justification for the sampling and how the data may be used.

Specific Comments:

Line 94: Wave glider, SWIFT, Saildrones, RAAVEN UAS, and SVPS acroynms all need to be defined. Perhaps they are commonly known in the NOAA community, I found it difficult to initially understand what these systems really were. Some are defined in the abstract, but it would be useful to define them all at the same point in the text, and this

seems a good place to do that.

Line 115: Change "Bridgetown" to "Bridgetown, Barbados".

Line 117: "abaft" is a jargon – and a better phrase is needed for a more general audience. What constitutes "high" concentrations? Are these periods when the instrument is sampling the ship's plume? If so, have these periods been masked out or defined with a flag in the dataset? In line 111, the text talks about avoiding contamination from the ship's stack, but the authors have not made clear in this sentence whether the peak values are contamination or not.

Figure 1: I found it very difficult to follow all the tracks in the two panels. The panel for leg 1 does not have the longitude defined (are they meant to be the same as the leg 2 panel). It would make more sense to me to have the insets as panels on the right side of the primary plots. It would be useful to include a larger geographic plot indicating the broader context of the study region. Perhaps a satellite image showing the clouds of interest in the region. Including island locations, such as Barbados is needed as well in the legs 1 and 2 panels.

Line 122: The paper focuses on data collected during legs 1 and 2. What is not clear is how many legs were there and over what time frame? Why focus on these legs as opposed to others (if there are any)? Some additional text is needed to clarify.

Figure 2: Please define the size range for the aerosol number concentration.

Line 133: This section describes the deployments of the various platforms during leg 1. It would be useful to start the section, with a brief description of the science objective associated with the sampling transects.

Line 224: as with the Section 2.1, it would be useful to state the objectives of this leg. Where the science objectives the same, and the objective was to simply collect more similar data? Was this a continuation of leg 1 and just a means of separating out the separate ship tracks?

Lines 229-230: Can you say what was the criteria for the SST fronts in determining where to deploy the SWIFTS?

Line 242: Was there a reason for the ship to remain at S3 for 3 days? Please define.

Lines 289-191: This sentence is a nice summary of the purpose of NTAS and its location. The sort of summary which is needed in parts previous two sections to define a purpose of the sampling strategy.

Lines 365-368: Given that one of the science objectives of ATOMIC was to study shallow oceanic convection, do the radar malfunctions affect the rest of the science that can be done with the dataset?

Line 376: It would be useful to provide an approximate height of the upper limit of valid data from the Doppler lidar.

Line 395: I do not see how the aerosol measurements on the ship reflect the objective of assessing the impact of aerosol-cloud interactions which occur within the clouds. Is the assumption that aerosols and CCN at the ocean surface are similar to those at cloud base? The radiation measurements will assess the column integrated impact of aerosols, but do not contain information on their vertical distribution and thus the possible concentrations at cloud level.

Line 414: What is meant by "reprocessed, gridded, and harmonized"? The next sentence implies a common vertical grid spacing for the radiosondes – so does this also mean a reprocessing and harmonization? Or is there something else meant by reprocessing and harmonization?

Line 663: Are there satellite measurements of SST that could be superimposed on Figure 1 maps to provide a context for the in situ measurements? The larger-scale image might be useful to go along with the description in this paragraph.

Line 821: Do you mean upwind and close to BACO? Seems that RHB was always upwind of BACO.

Line 826: Please include the measurement uncertainty value here.

Lines 828-830: Some plausible explanation for the difference in the aerosol size distribution is warranted. While the bimodal distribution is similar, the magnitude is larger at BACO. The previous paragraph implies CCN was similar between BACO and RHB during the same time period. CCN depends on aerosol size, but perhaps the differences in size do not have a significant impact at 0.4% supersaturation? Are the differences possibly due to differences in the instruments? What does this mean for scientists wishing to interpret aerosol-cloud interactions in the region?

---

## Referee Comment (RC2) · Anonymous Referee #2 · 4 Jan 2021

Review of "Measurements from the RV Ronald H. Brown and related platforms as part of the Atlantic Tradewind Ocean-Atmosphere Mesoscale Interaction Campaign (ATOMIC)" by Quinn et al.

This manuscript described comprehensive atmosphere and ocean data sets acquired the ATOMIC conducted from January to July 2020. The authors provide details of each measurement including sampling strategy, a overview, and inter-platform data comparisons. In situ measurements for various atmosphere and ocean variables on multi-platform are unique and very useful especially for air-sea interaction study in

addition to lower atmosphere and upper ocean studies. Inter-platform comparisons are also important to realize observation accuracy and uncertainty. Although there are detailed explanations for various measurements, several ambiguous points are also appeared. Thus, I suggest that the authors consider the following comments and then improve the manuscript.

Specific Comments:

Section 2:

- The nominal accuracy should be presented in tables (4-10).

- Available data period might be helpful for reader/user.

- The details for BCO measurements are also needed.

Section 3:

L659-661, L811-812: Figures of fires based on FIRMS associated with the events should be shown.

L700-701: Satellite-derived SSS and SST distributions should be shown to realize front position and structure.

L748: why the authors did not conduct height correction? L755-756: Is there any evidence? Do you have any insight from radiosonde measurements?

Section 4:

Sub-section 4. 3. 2.: I could not catch the point because there are relatively large differences between each comparison. What is the point from the comparisons? Need more explanation.

Minor comments:

- There are different expressions for "nautical miles". It should be unified.

[Figure]

- In sub-section 2. 11., abbreviations of CCN, DMA, DMT. . ..., should be explained.

- L808: "Jan. 9" might be "Feb.9".

---

## Author Comment (AC1) · 28 Jan 2021

For clarity, reviewers' comments are in *italics* below and responses are in normal type. Any change in wording from the original manuscript is indicated in red in the responses.

*Reviewer #1*

*General Comments:*

*1) The paper is well-written and organized and I just have some specific comments to clarify the details. At times, the paper uses nautical jargon that may be difficult for some atmospheric sciences to understand. The authors should try to make the text generic enough for both atmospheric and ocean science communities.*

*2) There are instances in which instrumentation did not work – which is common for field campaigns. But it would be useful for the authors to elaborate a bit on how the missing planned data might have on the overall science objectives of the campaign.*

*3) The paper does a good job at describing the how sampling was conducted during the campaign, but in many places it does not provide any rational as to why the sampling strategy was conducted to serve the objectives of the campaign. A few sentences here and there will help the reader understand the reason for those choices and provide potential users the scientific justification for the sampling and how the data may be used.*

All of the general comments have been addressed in response to the specific comments. Please see below.

*Specific Comments:*

*Line 94: Wave glider, SWIFT, Saildrones, RAAVEN UAS, and SVPS acroynms all need to be defined. Perhaps they are commonly known in the NOAA community, I found it difficult to initially understand what these systems really were. Some are defined in the abstract, but it would be useful to define them all at the same point in the text, and this seems a good place to do that.*

We have spelled out the acronym for RAAVEN in the abstract where all of the other acronyms are spelled out upon first use.

*Line 115: Change "Bridgetown" to "Bridgetown, Barbados".*

Done.

*Line 117: "abaft" is a jargon – and a better phrase is needed for a more general audience. What constitutes "high" concentrations? Are these periods when the instrument is sampling the ship's plume? If so, have these periods been masked out or defined with a flag in the dataset? In line 111, the text talks about avoiding contamination from*

*the ship's stack, but the authors have not made clear in this sentence whether the peak values are contamination or not.*

We have modified the text for a general audience and discussed how aerosol data collected during these periods were handled as follows:

"Periods of unfavorable winds for atmospheric sampling were identified by relative winds from behind the ship's beam (~ - 90° through 180° to + 90° relative to the bow at 0°). A time series of relative winds and corresponding high particle number concentrations due to emissions from the ship's stack (~> 1000 cm$^{-3}$) is shown in Figure 2. These periods have been removed from the aerosol data."

*Figure 1: I found it very difficult to follow all the tracks in the two panels. The panel for leg 1 does not have the longitude defined (are they meant to be the same as the leg 2 panel). It would make more sense to me to have the insets as panels on the right side of the primary plots. It would be useful to include a larger geographic plot indicating the broader context of the study region. Perhaps a satellite image showing the clouds of interest in the region. Including island locations, such as Barbados is needed as well in the legs 1 and 2 panels.*

Figure 1 has been changed as suggested. The insets are now separate plots on the right side. Figures 1a and b now have the island of Barbados labeled. Additional Caribbean islands also are shown.

*Line 122: The paper focuses on data collected during legs 1 and 2. What is not clear is how many legs were there and over what time frame? Why focus on these legs as opposed to others (if there are any)? Some additional text is needed to clarify.*

The following text has been added to the Introduction:

"ATOMIC was composed of two legs with Leg 1 conducted between Jan. 7 and 25, 2020 and Leg 2 conducted between Jan. 28 and Feb.13, 2020".

*Figure 2: Please define the size range for the aerosol number concentration.*

The following text has been added to the figure caption:

"…particle number concentration ($D_{gn} > 13$ nm)…"

*Line 133: This section describes the deployments of the various platforms during leg 1. It would be useful to start the section, with a brief description of the science objective associated with the sampling transects.*

The following text has been added to Section 2.1:

"During Leg 1, the NTAS mooring was swapped out, Wave Gliders were deployed for the duration of the experiment, and the SWIFTs were deployed and then recovered at the end of the leg. In addition to these

logistical operations, measurements were made throughout the leg to characterize atmospheric and oceanic conditions upwind of the EUREC[4]A study region."

In addition, we clarify with the following text that the ship went into port at the end of Leg 1:

"The ship ended Leg 1 with a transit around the southern end of Barbados and into Bridgetown with an arrival on Jan. 26 at 12:15 for open house and outreach activities to be conducted on Jan. 27."

*Line 224: as with the Section 2.1, it would be useful to state the objectives of this leg. Where the science objectives the same, and the objective was to simply collect more similar data? Was this a continuation of leg 1 and just a means of separating out the separate ship tracks?*

We have added the following text for clarification:

"During Leg 2, the SWIFTs were deployed at the beginning of the leg and then recovered along with the Wave Gliders at the end of the leg. Similar to Leg 1, measurements were made throughout the leg to characterize atmospheric and oceanic conditions upwind of the EUREC[4]A study region."

*Lines 229-230: Can you say what was the criteria for the SST fronts in determining where to deploy the SWIFTS?*

We have added the following text for clarification:

"The ship veered off its NE track on Jan. 29 at 20:18 and turned to the southeast to map the spatial orientation of SST fronts with gradients around 0.75°C for determining where to deploy SWIFTs."

*Line 242: Was there a reason for the ship to remain at S3 for 3 days? Please define.*

We have added the following text for clarification:
"The ship remained at S3 until Feb. 3 at 15:00 to characterize diurnal variations in oceanic and atmospheric conditions and to be in position for the P-3's RF5 and RF6. Continuous atmospheric and surface ocean measurements were made, radiosondes were launched every 4 hrs, and uCTD casts were conducted every 2 hrs."

*Lines 289-191: This sentence is a nice summary of the purpose of NTAS and its location. The sort of summary which is needed in parts previous two sections to define a purpose of the sampling strategy.*

Logistical and overarching goals for Legs 1 and 2 have been added in response to a previous comment.

*Lines 365-368: Given that one of the science objectives of ATOMIC was to study shallow oceanic convection, do the radar malfunctions affect the rest of the science that can be done with the dataset?*

We have added the following text for clarification:

"Although the loss of this information limited the ship-based based observations of non-precipitating cloud, data from the cloud radar on the P-3 will be used to fill in gaps."

*Line 376: It would be useful to provide an approximate height of the upper limit of valid data from the Doppler lidar.*

We have added the following sentence:

"The maximum height of valid data depends on the availability of aerosol scattering targets. Typically, the instrument provided data through the top of the marine atmospheric boundary layer, in the presence of elevated dust layers to 3km, and clouds to a height of 7km."

*Line 395: I do not see how the aerosol measurements on the ship reflect the objective of assessing the impact of aerosol-cloud interactions which occur within the clouds. Is the assumption that aerosols and CCN at the ocean surface are similar to those at cloud base? The radiation measurements will assess the column integrated impact of aerosols, but do not contain information on their vertical distribution and thus the possible concentrations at cloud level.*

The sentence has been changed as follows:

"The goals of NOAA's Pacific Marine Environmental Laboratory (PMEL) were to assess the impacts of aerosols on clouds and direct aerosol light scattering and absorption on the temporal variability of net radiation reaching the ocean surface and SST for the conditions of a well-mixed boundary layer."

*Line 414: What is meant by "reprocessed, gridded, and harmonized"? The next sentence implies a common vertical grid spacing for the radiosondes – so does this also mean a reprocessing and harmonization? Or is there something else meant by reprocessing and harmonization?*

The sentence has been changed as follows:

"The data were put into 10 m altitude bins and merged with the EUREC$^4$A sounding network."
As before, Stephan et al. (2020) is referenced for further details about post-processing.

*Line 663: Are there satellite measurements of SST that could be superimposed on Figure 1 maps to provide a context for the in situ measurements? The larger-scale image might be useful to go along with the description in this paragraph.*

The cruise tracks shown in Figure 1 are colored by SST and show the warmer temperatures near Barbados relative to the rest of the study region. Satellite images to shore up research-related results will be included in scientific papers that are in preparation.

*Line 821: Do you mean upwind and close to BACO? Seems that RHB was always upwind of BACO.*

The sentence has been changed as follows:

"The comparison when the ship was 20 NM upwind of BACO is indicated by the rectangle in Fig. 9."

*Line 826: Please include the measurement uncertainty value here.*

The sentence has been changed as follows:
"However, this difference is within the combined uncertainty of 30% for mono- and polydisperse CCN measurements."

*Lines 828-830: Some plausible explanation for the difference in the aerosol size distribution is warranted. While the bimodal distribution is similar, the magnitude is larger at BACO. The previous paragraph implies CCN was similar between BACO and RHB during the same time period. CCN depends on aerosol size, but perhaps the differences in size do not have a significant impact at 0.4% supersaturation? Are the differences possibly due to differences in the instruments? What does this mean for scientists wishing to interpret aerosol-cloud interactions in the region?*

Conjecture on sources of the difference in size distributions has been added as follows:
"Differences in magnitude could be due to instrumental issues or local aerosol sources at BACO."

*Reviewer #2*

*Specific Comments:*
*Section 2:*
*- The nominal accuracy should be presented in tables (4-10).*

Available measurement uncertainties are reported in the data set metadata. We have added a sentence as follows:

 "Available measurement uncertainties are reported in the data set metadata at ftp://ftp2.psl.noaa.gov/Projects/ATOMIC/data/ and https://www.ncei.noaa.gov/."

*- Available data period might be helpful for reader/user.*

The available data period is reported in Table 1 as "Sampling days". In addition, in response to reviewer 1 we added the following information to the introduction:

"ATOMIC was composed of two legs with Leg 1 conducted between Jan. 7 and 25, 2020 and Leg 2 conducted between Jan. 28 and Feb.13, 2020".

*- The details for BCO measurements are also needed.*

 A section (2.12) with a description of the BCO measurements has been added as follows:

 **"2.12. BCO measurements**

BCO is at a height of 25 m.a.s.l. Meteorological sensors (Vaisala WXT-520) were mounted at 4 m.a.g.l.  BCO launched 182 radiosondes. Data from the sondes were merged into the EUREC$^4$A sounding network (Stephan et al., 2020). A Lufft ceilometer CHN 15k NIMBUS was used for the determination of cloud base height."

*Section 3:*
*L659-661, L811-812: Figures of fires based on FIRMS associated with the events*
*should be shown.*

Given the number of figures already in the paper, we decline to add another one. Interested readers can visit the link that is provided. In addition, a figure from FIRMS will be provided in a research-oriented paper that is under preparation.

*L700-701: Satellite-derived SSS and SST distributions should be shown to realize front position and structure.*

SST along the cruise track are shown in Figure 1. Satellite images to shore up research-related results will be included in scientific papers that are in preparation.

*L748: why the authors did not conduct height correction?*

The changes for a height correction would be smaller than the uncertainties associated with adjusting overland values using surface fluxes from the *RV Ronald H. Brown.* We have added the following information:

 "BCO meteorological sensors were located at 30 m.a.s.l. and were not adjusted to a height of 10 m due to uncertainties in adjusting overland measurements (BC0) with surface fluxes from the *RV Ronald H. Brown.*"

*L755-756: Is there any evidence? Do you have any insight from radiosonde measurements?*

Unfortunately, we only have radiosondes from BCO during the inter-comparison period so it is not possible to compare to the ship.

*Section 4:*
*Sub-section 4. 3. 2.: I could not catch the point because there are relatively large differences between each comparison. What is the point from the comparisons? Need more explanation.*

The largest differences in cloud base height were between the RAAVEN and the other platforms during the RHB-BCO comparison. Since submitting the paper it was found that the RAAVEN sensors were not conditioned before flight. The RAAVEN data have now been corrected and the absolute average difference between the RHB lidar and the RAAVEN have decreased from 260 to 110 m (see Figure 11).

*Minor comments:*

*- There are different expressions for "nautical miles". It should be unified.*

Nautical miles is now "NM" throughout the paper.

*- In sub-section 2. 11., abbreviations of CCN, DMA, DMT. . ..., should be explained.*

Done.

*- L808: "Jan. 9" might be "Feb.9".*

 Yes, it is. Thank you!